# Incentivized Communication for Federated Bandits

**Zhepei Wei**[1*]   **Chuanhao Li**[1*]   **Haifeng Xu**[2]   **Hongning Wang**[1]
University of Virginia[1]     University of Chicago[2]
`{tqf5qb, cl5ev, hw5x}@virginia.edu`
`haifengxu@chicago.edu`

## Abstract

Most existing works on federated bandits take it for granted that all clients are altruistic about sharing their data with the server for the collective good whenever needed. Despite their compelling theoretical guarantee on performance and communication efficiency, this assumption is overly idealistic and oftentimes violated in practice, especially when the algorithm is operated over self-interested clients, who are reluctant to share data without explicit benefits. Negligence of such self-interested behaviors can significantly affect the learning efficiency and even the practical operability of federated bandit learning. In light of this, we aim to spark new insights into this under-explored research area by formally introducing an incentivized communication problem for federated bandits, where the server shall motivate clients to share data by providing incentives. Without loss of generality, we instantiate this bandit problem with the contextual linear setting and propose the first incentivized communication protocol, namely, INC-FEDUCB, that achieves near-optimal regret with provable communication and incentive cost guarantees. Extensive empirical experiments on both synthetic and real-world datasets further validate the effectiveness of the proposed method across various environments.

## 1   Introduction

Federated bandit learning has recently emerged as a promising new direction to promote the application of bandit models while preserving privacy by enabling collaboration among multiple distributed clients [10, 40, 22, 13, 23, 24, 25, 15, 37, 9]. The main focus in this line of research is on devising communication-efficient protocols to achieve near-optimal regret in various settings. Most notably, the direction on federated contextual bandits has been actively gaining momentum, since the debut of several benchmark communication protocols for contextual linear bandits in the P2P [18] and star-shaped [40] networks. Many subsequent studies have explored diverse configurations of the clients' and environmental modeling factors and addressed new challenges arising in these contexts. Notable recent advancements include extensions to asynchronous linear bandits [22, 13], generalized liner bandits [23], and kernelized contextual bandits [24, 25].

Despite the extensive exploration of various settings, almost all existing federated bandit algorithms rely on the assumption that every client in the system is willing to share their local data/model with the server, regardless of the communication protocol design. For instance, synchronous protocols [40, 23] require all clients to simultaneously engage in data exchange with the server in every communication round. Similarly, asynchronous protocols [22, 25, 13] also assume clients must participate in communication as long as the individualized upload or download event is triggered, albeit allowing interruptions by external factors (e.g., network failure).

In contrast, our work is motivated by the practical observation that many clients in a federated system are inherently self-interested and thus reluctant to share data without receiving explicit benefits from the server [16]. For instance, consider the following scenario: a recommendation platform

---

[*]Equal Contribution

37th Conference on Neural Information Processing Systems (NeurIPS 2023).

(server) wants its mobile app users (clients) to opt in its new recommendation service, which switches previous on-device local bandit algorithm to a federated bandit algorithm. Although the new service is expected to improve the overall recommendation quality for all clients, particular clients may not be willing to participate in this collaborative learning, as the expected gain for them might not compensate their locally increased cost (e.g., communication bandwidth, added computation, lost control of their data, and etc). In this case, additional actions have to be taken by the server to encourage participation, as it has no power to force clients. This exemplifies the most critical concern in the real-world application of federated learning [16]. And a typical solution is known as *incentive mechanism*, which motivates individuals to contribute to the social welfare goal by offering incentives such as monetary compensation.

While recent studies have explored incentivized data sharing in federated learning [30, 38], most of which only focused on the supervised offline learning setting [16]. To our best knowledge, ours is the first work that studies incentive design for federated bandit learning, which inherently imposes new challenges. First, there is a lack of well-defined metric to measure the utility of data sharing, which rationalizes a client's participation. Under the context of bandit learning, we measure data utility by the expected regret reduction from the exchanged data for each client. As a result, each client values data (e.g., sufficient statistics) from the server differently, depending on how such data aligns with their local data (e.g., the more similar the less valuable). Second, the server is set to minimize regret across all clients through data exchange. But as the server does not generate data, it can be easily trapped by the situation where its collected data cannot pass the critical mass to ensure every participating client's regret is close to optimal (e.g., the data under server's possession cannot motivate the clients who have more valuable data to participate). To break the deadlock, we equip the server to provide monetary incentives. Subsequently, the server needs to minimize its cumulative monetary payments, in addition to the regret and communication minimization objectives as required by federated bandit learning. We propose a provably effective incentivized communication protocol, based on a heuristic search strategy to balance these distinct learning objectives. Our solution obtains near-optimal regret $O(d\sqrt{T}\log T)$ with provable communication and incentive cost guarantees. Extensive empirical simulations on both synthetic and real-world datasets further demonstrate the effectiveness of the proposed protocol in various federated bandit learning environments.

## 2   Related Work

**Federated Bandit Learning**    One important branch in this area is federated multi-armed bandits (MABs), which has been well-studied in the literature [27, 36, 19, 4, 20, 28, 32, 39, 34, 33, 43]. The other line of work focuses on the federated contextual bandit setting [18, 40], which has recently attracted increasing attention. Wang et al. [40] and Korda et al. [18] are among the first to investigate this problem, where multiple communication protocols for linear bandits [1, 26] in star-shaped and P2P networks are proposed. Many follow-up works on federated linear bandits [10, 15, 22, 13] have emerged with different client and environment settings, such as investigating fixed arm set [15], incorporating differential privacy [10], and introducing asynchronous communication [13, 22]. Li et al. [23] extend the federated linear bandits to generalized linear bandits [11]. And they further investigated federated learning for kernelized contextual bandits in both synchronous and asynchronous settings [24, 25].

In this work, we situate the incentivized federated bandit learning problem under linear bandits with time-varying arm sets, which is a popular setting in many recent works [40, 10, 22, 13]. But we do not assume the clients will always participate in data sharing: they will choose not to share their data with the server if the resulting benefit of data sharing is not deemed to outweigh the cost. Here we need to differentiate our setting from those with asynchronous communication, e.g., Asyn-LinUCB [22]. Such algorithms still assume all clients are willing to share, though sometimes the communication can be interrupted by some external factors (e.g., network failure). We do not assume communication failures and leave it as our future work. Instead, we assume the clients need to be motivated to participate in federated learning, and our focus is to devise the minimum incentives to obtain the desired regret and communication cost for all participating clients.

**Incentivized Federated Learning**    Data sharing is essential to the success of federated learning [30], where client participation plays a crucial role. However, participation involves costs, such as the need for additional computing and communication resources, and the risk of potential privacy breaches, which can lead to opt-outs [5, 14]. In light of this, recent research has focused on investigating

incentive mechanisms that motivate clients to contribute, rather than assuming their willingness to participate. Most of the existing research involves multiple decentralized clients solving the same task, typically with different copies of IID datasets, where the focus is on designing data valuation methods that ensure fairness or achieve a specific accuracy objective [35, 41, 8]. On the other hand, Donahue et al. [7] study voluntary participation in model-sharing games, where clients may opt out due to biased global models caused by the aggregated non-IID datasets. More recently, Karimireddy et al. [16] investigated incentive mechanism design for data maximization while avoiding free riders. For a detailed discussion of this topic, we refer readers to recent surveys on incentive mechanism design in federated learning [42, 38].

However, most works on incentivized federated learning only focus on better model estimation among fixed offline datasets, which does not apply to the bandit learning problem, where the exploration of growing data is also part of the objective. More importantly, in our incentivized federated bandit problem, the server is obligated to improve the overall performance of the learning system, i.e., minimizing regret among all clients, which is essentially different from previous studies where the server only selectively incentivizes clients to achieve a certain accuracy [35] or to investigate how much accuracy the system can achieve without payment [16].

## 3 Preliminaries

In this section, we formally introduce the incentivized communication problem for federated bandits under the contextual linear bandit setting.

### 3.1 Federated Bandit Learning

We consider a learning system consisting of (1) $N$ clients that directly interact with the environment by taking actions and receiving the corresponding rewards, and (2) a central server that coordinates the communication among the clients to facilitate their learning collectively. The clients can only communicate with the central server, but not with each other, resulting in a star-shaped communication network. At each time step $t \in [T]$, an arbitrary client $i_t \in [N]$ becomes active and chooses an arm $\mathbf{x}_t$ from a candidate set $\mathcal{A}_t \subseteq \mathbb{R}^d$, and then receives the corresponding reward feedback $y_t = f(\mathbf{x}_t) + \eta_t \in \mathbb{R}$. Note that $\mathcal{A}_t$ is time-varying, $f$ denotes the unknown reward function shared by all clients, and $\eta_t$ denotes zero mean sub-Gaussian noise with known variance $\sigma^2$.

The performance of the learning system is measured by the cumulative (pseudo) regret over all $N$ clients in the finite time horizon $T$, i.e., $R_T = \sum_{t=1}^{T} r_t$, where $r_t = \max_{\mathbf{x} \in \mathcal{A}_t} \mathbf{E}[y|\mathbf{x}] - \mathbf{E}[y_t|\mathbf{x}_t]$ is the regret incurred by client $i_t$ at time step $t$. Moreover, under the federated learning setting, the system also needs to keep the communication cost $C_T$ low, which is measured by the *total number of scalars* [40] being transferred across the system up to time $T$. With the linear reward assumption, i.e., $f(\mathbf{x}) = \mathbf{x}^\top \theta_\star$, where $\theta_\star$ denotes the unknown parameter, a ridge regression estimator $\hat{\theta}_t = V_t^{-1} b_t$ can be constructed based on sufficient statistics from all $N$ clients at each time step $t$, where $V_t = \sum_{s=1}^{t} \mathbf{x}_s \mathbf{x}_s^\top$ and $b_t = \sum_{s=1}^{t} \mathbf{x}_s y_s$ [21]. Using $\hat{\theta}_t$ under the Optimism in the Face of Uncertainty (OFUL) principle [1], one can obtain the optimal regret $R_T = O(d\sqrt{T})$. To achieve this regret bound in the federated setting, a naive method is to immediately share statistics of each newly collected data sample to all other clients in the system, which essentially recovers its centralized counterpart. However, this solution incurs a disastrous communication cost $C_T = O(d^2 NT)$. On the other extreme, if no communication occurs throughout the entire time horizon (i.e., $C_T = 0$), the regret upper bound can be up to $R_T = O(d\sqrt{NT})$ when each client interacts with the environment at the same frequency, indicating the importance of timely data/model aggregation in reducing $R_T$.

To balance this trade-off between regret and communication cost, prior research efforts centered around designing communication-efficient protocols for federated bandits that feature the "delayed update" of sufficient statistics [40, 22, 13]. Specifically, each client $i$ only has a delayed copy of $V_t$ and $b_t$, denoted as $V_{i,t} = V_{t_{\text{last}}} + \Delta V_{i,t}, b_{i,t} = b_{t_{\text{last}}} + \Delta b_{i,t}$, where $V_{t_{\text{last}}}, b_{t_{\text{last}}}$ is the aggregated sufficient statistics shared by the server in the last communication, and $\Delta V_{i,t}, \Delta b_{i,t}$ is the accumulated local updates that client $i$ obtain from its interactions with the environment since $t_{\text{last}}$. In essence, the success of these algorithms lies in the fact that $V_t, b_t$ typically changes slowly and thus has little instant impact on the regret for most time steps. Therefore, existing protocols that only require occasional communications can still achieve nearly optimal regret, despite the limitation on assuming clients' willingness on participation as we discussed before.

## 3.2 Incentivized Federated Bandits

Different from the prior works in this line of research, where all clients altruistically share their data with the server whenever a communication round is triggered, we are intrigued in a more realistic setting where clients are *self-interested* and thus reluctant to share data with the server if not well motivated. Formally, each client in the federated system inherently experiences a cost[2] of data sharing, denoted by $\widetilde{D}_i^p \in \mathbb{R}$, due to their individual consumption of computing resources in local updates or concerns about potential privacy breaches caused by communication with the server. Moreover, as the client has nothing to lose when there is no local update to share in a communication round at time step $t$, in this case we assume the cost is 0, i.e., $D_i^p = \widetilde{D}_i^p \cdot \mathbb{I}(\Delta V_{i,t} \neq \mathbf{0})$. As a result, the server needs to motivate clients to participate in data sharing via the incentive mechanism $\mathcal{M} : \mathbb{R}^N \times \mathbb{R}^{d \times d} \to \mathbb{R}^N$, which takes as inputs a collection of client local updates $\Delta V_{i,t} \in \mathbb{R}^{d \times d}$ and a vector of cost values $D^p = \{D_1^p, \cdots, D_N^p\} \in \mathbb{R}^N$, and outputs the incentive $\mathcal{I} = \{\mathcal{I}_{1,t}, \cdots, \mathcal{I}_{N,t}\} \in \mathbb{R}^N$ to be distributed among the clients. Specifically, to make it possible to measure gains and losses of utility in terms of real-valued incentives (e.g., monetary payment), we adopt the standard *quasi-linear* utility function assumption, as is standard in economic analysis [2, 31].

At each communication round, a client decides whether to share its local update with the server based on the potential utility gained from participation, i.e., the difference between the incentive and the cost of data sharing. This requires the incentive mechanism to be *individually rational*:

**Definition 1 (Individual Rationality [29])** *An incentive mechanism* $\mathcal{M} : \mathbb{R}^N \times \mathbb{R}^{d \times d} \to \mathbb{R}^N$ *is individually rational if for any $i$ in the participant set $S_t$ at time step $t$, we have*

$$\mathcal{I}_{i,t} \geq D_i^p \tag{1}$$

*In other words, each participant must be guaranteed non-negative utility by participating in data sharing under $\mathcal{M}$.*

The server coordinates with all clients and incentivizes them to participate in the communication to realize its own objective (e.g., collective regret minimization). This requires $\mathcal{M}$ to be *sufficient*:

**Definition 2 (Sufficiency)** *An incentive mechanism* $\mathcal{M} : \mathbb{R}^N \times \mathbb{R}^{d \times d} \to \mathbb{R}^N$ *is sufficient if the resulting outcome satisfies the server's objective.*

Typically, under different application scenarios, the server may have different objectives, such as regret minimization or best arm identification. In this work, we set the objective of the server to minimize the regret across all clients; and ideally the server aims to attain the optimal $\widetilde{O}(d\sqrt{T})$ regret in the centralized setting via the incentivized communication. Therefore, we consider an incentive mechanism is sufficient if it ensures that the resulting accumulated regret is bounded by $\widetilde{O}(d\sqrt{T})$.

## 4 Methodology

The communication backbone of our solution derives from DisLinUCB [40], which is a widely adopted paradigm for federated linear bandits. We adopt their strategy for arm selection and communication trigger, so as to focus on the incentive mechanism design. We name the resulting algorithm INC-FEDUCB, and present it in Algorithm 1. Note that the two incentive mechanisms to be presented in Section 4.2 and 4.3 are not specific to any federated bandit learning algorithms, and each of them can be easily extended to alternative workarounds as a plug-in to accommodate the incentivized federated learning setting. For clarity, a summary of technical notations can be found in Table 7.

### 4.1 A General Framework: INC-FEDUCB Algorithm

Our framework comprises three main steps: 1) client's local update; 2) communication trigger; and 3) incentivized data exchange among the server and clients. Specifically, after initialization, an active client performs a local update in each time step and checks the communication trigger. If a communication round is triggered, the system performs incentivized data exchange between clients and the server. Otherwise, no communication is needed.

---

[2]Note that if the costs are trivially set to zero, then clients have no reason to opt-out of data sharing and our problem essentially reduces to the standard federated bandits problem [40].

**Algorithm 1** INC-FEDUCB Algorithm

---

**Require:** $D_c \geq 0$, $D^p = \{D_1^p, \cdots, D_N^p\}$, $\sigma$, $\lambda > 0$, $\delta \in (0, 1)$

1: Initialize: **[Server]** $V_{g,0} = \mathbf{0}_{d \times d} \in \mathbb{R}^{d \times d}$, $b_{g,0} = \mathbf{0}_d \in \mathbb{R}^d$
2:                 $\Delta V_{-j,0} = \mathbf{0}_{d \times d}$, $\Delta b_{-j,0} = \mathbf{0}_d$, $\forall j \in [N]$
3:         **[All clients]** $V_{i,0} = \mathbf{0}_{d \times d}$, $b_{i,0} = \mathbf{0}_d$, $\Delta V_{i,0} = \mathbf{0}_{d \times d}$, $\Delta b_{i,0} = \mathbf{0}_d$, $\Delta t_{i,0} = 0$, $\forall i \in [N]$
4: **for** $t = 1, 2, \ldots, T$ **do**
5:      **[Client $i_t$]** Observe arm set $\mathcal{A}_t$
6:      **[Client $i_t$]** Select arm $\mathbf{x}_t \in \mathcal{A}_t$ by Eq. (2) and observe reward $y_t$
7:      **[Client $i_t$]** Update: $V_{i_t,t} \mathrel{+}= \mathbf{x}_t \mathbf{x}_t^\top$, $b_{i_t,t} \mathrel{+}= \mathbf{x}_t y_t$
8:               $\Delta V_{i_t,t} \mathrel{+}= \mathbf{x}_t \mathbf{x}_t^\top$, $\Delta b_{i_t,t} \mathrel{+}= \mathbf{x}_t y_t$, $\Delta t_{i_t,t} \mathrel{+}= 1$
9:      **if** $\Delta t_{i_t,t} \log \frac{\det(V_{i_t,t}+\lambda I)}{\det(V_{i_t,t} - \Delta V_{i_t,t}+\lambda I)} > D_c$ **then**
10:         **[All clients $\rightarrow$ Server]** Upload $\Delta V_{i,t}$ such that $\widetilde{S}_t = \{\Delta V_{i,t} | \forall i \in [N]\}$
11:         **[Server]** Select incentivized participants $S_t = \mathcal{M}(\tilde{S}_t)$         ▷ Incentive Mechanism
12:         **for** $i : \Delta V_{i,t} \in S_t$ **do**
13:             **[Participant $i \rightarrow$ Server]** Upload $\Delta b_{i,t}$
14:             **[Server]** Update: $V_{g,t} \mathrel{+}= \Delta V_{i,t}$, $b_{g,t} \mathrel{+}= \Delta b_{i,t}$
15:                  $\Delta V_{-j,t} \mathrel{+}= \Delta V_{i,t}$, $\Delta b_{-j,t} \mathrel{+}= \Delta b_{i,t}$, $\forall j \neq i$
16:             **[Participant $i$]** Update: $\Delta V_{i,t} = 0$, $\Delta b_{i,t} = 0$, $\Delta t_{i,t} = 0$
17:         **for** $\forall i \in [N]$ **do**
18:             **[Server $\rightarrow$ All Clients]** Download $\Delta V_{-i,t}$, $\Delta b_{-i,t}$
19:             **[Client $i$]** Update: $V_{i,t} \mathrel{+}= \Delta V_{-i,t}$, $b_{i,t} \mathrel{+}= \Delta b_{-i,t}$
20:             **[Server]** Update: $\Delta V_{-i,t} = 0$, $\Delta b_{-i,t} = 0$

---

Formally, at each time step $t = 1, \ldots, T$, an arbitrary client $i_t$ becomes active and interacts with its environment using observed arm set $\mathcal{A}_t$ (Line 5). Specifically, it selects an arm $\mathbf{x}_t \in \mathcal{A}_t$ that maximizes the UCB score as follows (Line 6):

$$\mathbf{x}_t = \arg\max_{\mathbf{x} \in \mathcal{A}_t} \mathbf{x}^\top \hat{\theta}_{i_t,t-1}(\lambda) + \alpha_{i_t,t-1} ||\mathbf{x}||_{V_{i_t,t-1}^{-1}(\lambda)} \tag{2}$$

where $\hat{\theta}_{i_t,t-1}(\lambda) = V_{i_t,t-1}^{-1}(\lambda) b_{i_t,t-1}$ is the ridge regression estimator of $\theta_\star$ with regularization parameter $\lambda > 0$, $V_{i_t,t-1}(\lambda) = V_{i_t,t-1} + \lambda I$, and $\alpha_{i_t,t-1} = \sigma \sqrt{\log \frac{\det(V_{i_t,t-1}(\lambda))}{\det(\lambda I)} + 2\log 1/\delta} + \sqrt{\lambda}$. $V_{i_t,t}(\lambda)$ denotes the covariance matrix constructed using data available to client $i_t$ up to time $t$. After obtaining a new data point $(\mathbf{x}_t, y_t)$ from the environment, client $i_t$ checks the communication event trigger $\Delta t_{i_t,t} \cdot \log \frac{\det(V_{i_t,t}(\lambda))}{\det(V_{i_t,t_{\text{last}}}(\lambda))} > D_c$ (Line 9), where $\Delta t_{i_t,t}$ denotes the time elapsed since the last time $t_{\text{last}}$ it communicated with the server and $D_c \geq 0$ denotes the specified threshold.

**Incentivized Data Exchange**    With the above event trigger, communication rounds only occur if (1) a substantial amount of new data has been accumulated locally at client $i_t$, and/or (2) significant time has elapsed since the last communication. However, in our incentivized setting, triggering a communication round does not necessarily lead to data exchange at time step $t$, as the participant set $S_t$ may be empty (Line 11). This characterizes the fundamental difference between INC-FEDUCB and DisLinUCB [40]: we no longer assume all $N$ clients will share their data with the server in an altruistic manner; instead, a rational client only shares its local update with the server if the condition in Eq. (1) is met. In light of this, to evaluate the potential benefit of data sharing, all clients must first reveal the value of their data to the server before the server determines the incentive. Hence, after a communication round is triggered, all clients upload their latest sufficient statistics update $\Delta V_{i,t}$ to the server (Line 10) to facilitate data valuation and participant selection in the incentive mechanism (Line 11). Note that this disclosure does not compromise clients' privacy, as the clients' secret lies in $\Delta b_{i,t}$ that is constructed by the rewards. Only participating clients will upload their $\Delta b_{i,t}$ to the server (Line 13). After collecting data from all participants, the server downloads the aggregated updates $\Delta V_{-i,t}$ and $\Delta b_{-i,t}$ to every client $i$ (Line 17-20). Following the convention in federated bandit learning [40], the communication cost is defined as the total number of scalars transferred during this data exchange process.

---

**Algorithm 2** Payment-free Incentive Mechanism

---

**Require:** $D^p = \{D_i^p | i \in [N]\}, \widetilde{S}_t = \{\Delta V_{i,t} | i \in [N]\}$

  1:  Initialize participant set $S_t = \widetilde{S}_t$
  2:  **while** $S_t \neq \emptyset$ **do**                 ▷ iteratively update $S_t$ until it becomes stable
  3:      StableFlag = True
  4:      **for** $i : \Delta V_{i,t} \in S_t$ **do**
  5:          **if** $\mathcal{I}_{i,t} < D_i^p$ **then**                            ▷ Eq. 4
  6:             Update participant set $S_t = S_t \setminus \{\Delta V_{i,t}\}$        ▷ remove client $j$ from $S_t$
  7:             StableFlag = False
  8:             **break**
  9:      **if** StableFlag = True **then**
 10:          **break**
 11:  **return** $S_t \subseteq \widetilde{S}_t$

---

### 4.2 Payment-free Incentive Mechanism

As mentioned in Section 1, in federated bandit learning, clients can reduce their regret by using models constructed via shared data. Denote $\widetilde{V}_t$ as the covariance matrix constructed by all available data in the system at time step $t$. Based on Lemma 5 and 7, the instantaneous regret of client $i_t$ is upper bounded by:

$$r_t \leq 2\alpha_{i_t, t-1} \sqrt{\mathbf{x}_t^\top \widetilde{V}_{t-1}^{-1} \mathbf{x}_t} \cdot \sqrt{\frac{\det(\widetilde{V}_{t-1})}{\det(V_{i_t, t-1})}} = O\left(\sqrt{d \log \frac{T}{\delta}}\right) \cdot \|\mathbf{x}_t\|_{\widetilde{V}_{t-1}^{-1}} \cdot \sqrt{\frac{\det(\widetilde{V}_{t-1})}{\det(V_{i_t, t-1})}} \quad (3)$$

where the determinant ratio reflects the additional regret due to the delayed synchronization between client $i_t$'s local sufficient statistics and the global optimal oracle. Therefore, **minimizing this ratio directly corresponds to reducing client $i_t$'s regret**. For example, full communication keeps the ratio at 1, which recovers the regret of the centralized setting discussed in Section 3.1.

Therefore, given the client's desire for regret minimization, the data itself can be used as a form of incentive by the server. And the star-shaped communication network also gives the server an information advantage over any single client in the system: a client can only communicate with the server, while the server can communicate with every client. Therefore, the server should utilize this advantage to create incentives (i.e., the LHS of Eq. (1)), and a natural design to evaluate this data incentive is:

$$\mathcal{I}_{i,t} := \mathcal{I}_{i,t}^d = \frac{\det(D_{i,t}(S_t) + V_{i,t})}{\det(V_{i,t})} - 1. \quad (4)$$

where $D_{i,t}(S_t) = \sum_{j:\{\Delta V_{j,t} \in S_t\} \wedge \{j \neq i\}} \Delta V_{j,t} + \Delta V_{-i,t}$ denotes the data that the server can offer to client $i$ during the communication at time $t$ (i.e., current local updates from other participants that have not been shared with the server) and $\Delta V_{-i,t}$ is the historically aggregated updates stored in the server that has not been shared with client $i$. Eq. (4) suggests a substantial increase in the determinant of the client's local data is desired by the client, which ultimately results in regret reduction.

With the above data valuation in Eq. (4), we propose the *payment-free* incentive mechanism that motivates clients to share data by redistributing data collected from participating clients. We present this mechanism in Algorithm 2, and briefly sketch it below. First, we initiate the participant set $S_t = \widetilde{S}_t$, assuming all clients agree to participate. Then, we iteratively update $S_t$ by checking the willingness of each client $i$ in $S_t$ according to Eq. (1). If $S_t$ is empty or all clients in it are participating, then terminate; otherwise, remove client $i$ from $S_t$ and repeat the process.

While this payment-free incentive mechanism is neat and intuitive, it has no guarantee on the amount of data that can be collected. To see this, we provide a theoretical negative result with rigorous regret analysis in Theorem 3 (see proof in Appendix C).

**Theorem 3 (Sub-optimal Regret)** *When there are at most $\frac{c}{2C} \frac{N}{\log(T/N)}$ number of clients (for some constant $C, c > 0$), whose cost value $D_i^p \leq \min\{(1 + \frac{L^2}{\lambda})^T, (1 + \frac{TL^2}{\lambda d})^d\}$, there exists a linear bandit instance with $\sigma = L = S = 1$ such that for $T \geq Nd$, the expected regret for* INC-FEDUCB *algorithm with payment-free incentive mechanism is at least $\Omega(d\sqrt{NT})$.*

**Algorithm 3** Payment-efficient Incentive Mechanism

---

**Require:** $\widetilde{S}_t = \{\Delta V_{i,t} | i \in [N]\}$, data-incentivized participant set $\widehat{S}_t \subseteq \widetilde{S}_t$, threshold $\beta$

1: **for** client $i : \Delta V_{i,t} \in \widetilde{S}_t \setminus \widehat{S}_t$ **do:**
2:      Compute client's potential contribution to the server (i.e., marginal gain in determinant):

$$c_{i,t}(\widehat{S}_t) = \det(\Delta V_{i,t} + V_{g,t}(\widehat{S}_t))/\det(V_{g,t}(\widehat{S}_t)), \;\; V_{g,t}(S_t) = V_{g,t-1} + \Sigma(S_t) \quad (6)$$

3:   Rank clients $\{i_1, \ldots, i_m\}$ by their potential contribution, where $m = |\widetilde{S}_t \setminus \widehat{S}_t|$
4:   Segment the list by finding $\alpha = \min\{j \mid \frac{\det(V_{g,t}(\widehat{S}_t) + \Delta V_{i_j,t})}{\det(V_{g,t}(\widetilde{S}_t))} \geq \beta, \; \forall j \in [m]\}$
5:   $k = \alpha - 1, \mathcal{I}_{\text{last}}^m = D_{i_\alpha}^p - \mathcal{I}_{i_\alpha,t}^d$
6:   **return** participant set $S_t = Heuristic\ Search(k, \mathcal{I}_{\text{last}}^m)$         ▷ Algorithm 4

---

Recall the discussion in Section 3.1, when there is no communication $R_T$ is upper bounded by $O(d\sqrt{NT})$. Hence, in the worst-case scenario, the payment-free incentive mechanism might not motivate any client to participate. It is thus not a sufficient mechanism.

### 4.3 Payment-efficient Incentive Mechanism

To address the insufficiency issue, we further devise a *payment-efficient* incentive mechanism that introduces additional monetary incentives to motivate clients' participation:

$$\mathcal{I}_{i,t} := \mathcal{I}_{i,t}^d + \mathcal{I}_{i,t}^m \quad (5)$$

where $\mathcal{I}_{i,t}^d$ is the data incentive defined in Eq. (4), and $\mathcal{I}_{i,t}^m$ is the real-valued monetary incentive, i.e., the payment assigned to the client for its participation. Specifically, we are intrigued by the question: rather than trivially paying unlimited amounts to ensure everyone's participation, can we devise an incentive mechanism that guarantees a certain level of client participation such that the overall regret is still nearly optimal but under acceptable monetary incentive cost?

Inspired by the determinant ratio principle discussed in Eq. (3), we propose to control the overall regret by ensuring that every client closely approximates the oracle after each communication round, which can be formalized as $\det(V_{g,t})/\det(\widetilde{V}_t) \geq \beta$, where $V_{g,t} = V_{g,t-1} + \Sigma(S_t)$ is to be shared with all clients and $\Sigma(S_t) = \sum_{j:\{\Delta V_{j,t} \in S_t\}} \Delta V_{j,t}$. The parameter $\beta \in [0, 1]$ characterizes the chosen gap between the practical and optimal regrets that the server commits to. Denote the set of clients motivated by $\mathcal{I}_{i,t}^d$ at time $t$ as $S_t^d$ and those motivated by $\mathcal{I}_{i,t}^m$ as $S_t^m$, and thus $S_t = S_t^m \cup S_t^d$. At each communication round, the server needs to find the minimum $\mathcal{I}_{i,t}^m$ such that pooling local updates from $S_t$ satisfies the required regret reduction for the entire system.

Note that Algorithm 2 maximizes $\mathcal{I}_{i,t}^d$, and thus the servers should compute $\mathcal{I}_{i,t}^m$ on top of optimal $\mathcal{I}_{i,t}^d$ and resulting $S_t^d$, which however is still combinatorially hard. First, a brute-force search can yield a time complexity up to $O(2^N)$. Second, different from typical optimal subset selection problems [17], the dynamic interplay among clients in our specific context brings a unique challenge: once a client is incentivized to share data, the other uninvolved clients may change their willingness due to the increased data incentive, making the problem even more intricate.

To solve the above problem, we propose a heuristic ranking-based method, as outlined in Algorithm 3. The heuristic is to rank clients by the marginal gain they bring to the server's determinant, as formally defined in Eq. (6), which helps minimize the number of clients requiring monetary incentives, while empowering the participation of other clients motivated by the aggregated data. This forms an iterative search process: First, we rank all $m$ non-participating clients (Line 2-3) by their potential contribution to the server (with participant set $S_t$ committed); Then, we segment the list by $\beta$, anyone whose participation satisfies the overall $\beta$ gap constraint is an immediately valid choice (Line 4). The first client $i_\alpha$ in the valid list and its payment $\mathcal{I}_{\text{last}}^m$ ($\infty$ if not available) will be our *last resort* (Line 5); Lastly, we check if there exist potentially more favorable solutions from the invalid list (Line 6). Specifically, we try to elicit up to $k = \alpha - 1$ ($k = m$ if $i_\alpha$ is not available) clients from the invalid list in $n \leq k$ rounds, where only one client will be chosen using the same heuristic in each round. If having $n$ clients from the invalid list also satisfies the $\beta$ constraint and results in a reduced monetary

incentive cost compared to $\mathcal{I}_{\text{last}}^m$, then we opt for this alternative solution. Otherwise, we will adhere to the *last resort*.

This *Heuristic Search* is detailed in Appendix A, and it demonstrates a time complexity of only $O(N)$ in the worst-case scenarios, i.e., $n = m = N$. Theorem 4 guarantees the sufficiency of this mechanism *w.r.t* communication and payment bounds.

**Theorem 4** *Under threshold $\beta$ and clients' committed data sharing cost $D^p = \{D_1^p, \cdots, D_N^p\}$, with high probability the monetary incentive cost of* INC-FEDUCB *satisfies*

$$M_T = O\left(\max D^p \cdot P \cdot N - \sum_{i=1}^{N} P_i \cdot \left(\frac{\det(\lambda I)}{\det(V_T)}\right)^{\frac{1}{P_i}}\right).$$

*where $P_i$ is the number of epochs client $i$ gets paid throughout time horizon $T$, $P$ is the total number of epochs, which is bounded $P = O(Nd \log T)$ by setting communication threshold $D_c = \frac{T}{N^2 d \log T} - \sqrt{\frac{T^2}{N^2 dR \log T}} \log \beta$, where $R = \lceil d \log(1 + \frac{T}{\lambda d}) \rceil$.*

*Henceforth, the communication cost satisfies*

$$C_T = O(Nd^2) \cdot P = O(N^2 d^3 \log T)$$

*Furthermore, by setting $\beta \geq e^{-\frac{1}{N}}$, the cumulative regret is*

$$R_T = O\left(d\sqrt{T} \log T\right)$$

The proof of theorem 4 can be found in Appendix D.

## 5 Experiments

We simulate the incentivized federated bandit problem under various environment settings. Specifically, we create an environment of $N = 50$ clients with cost of data sharing $D^p = \{D_1^p, \cdots, D_N^p\}$, total number of iterations $T = 5,000$, feature dimension $d = 25$, and time-varing arm pool size $K = 25$. By default, we set $D_i^p = D_\star^p \in \mathbb{R}, \forall i \in [N]$. Due to the space limit, more detailed results and discussions on real-world dataset can be found in Appendix E.

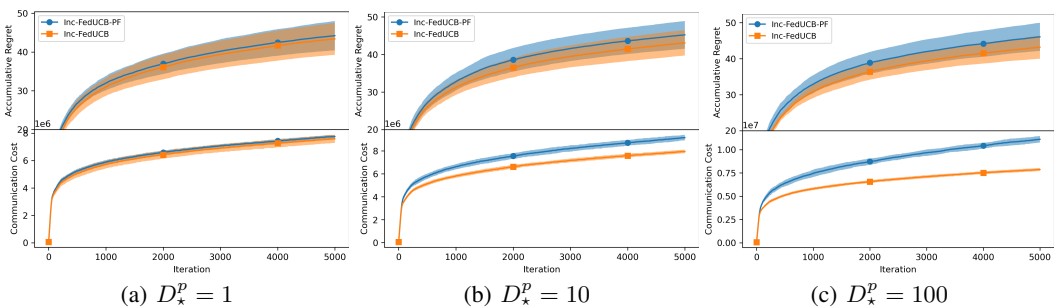

(a) $D_\star^p = 1$        (b) $D_\star^p = 10$        (c) $D_\star^p = 100$

Figure 1: Comparison between payment-free vs. payment-efficient incentive designs. The results are averaged over 10 runs with standard deviation as the error bars.

### 5.1 Payment-free vs. Payment-efficient

We first empirically compared the performance of the payment-free mechanism (named as INC-FEDUCB-PF) and the payment-efficient mechanism INC-FEDUCB in Figure 1. It is clear that the added monetary incentives lead to lower regret and communication costs, particularly with increased $D_\star^p$. Lower regret is expected as more data can be collected and shared; while the reduced communication cost is contributed by reduced communication frequency. When less clients can be motivated in one communication round, more communication rounds will be triggered as the clients tend to have outdated local statistics.

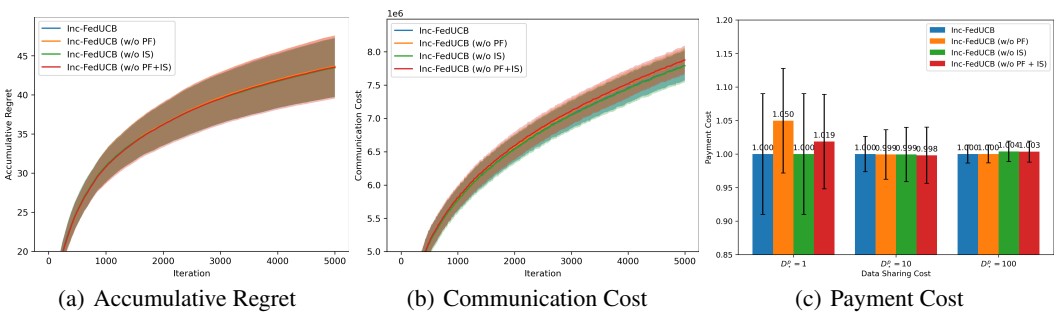

|   (a) Accumulative Regret   |   (b) Communication Cost   |   (c) Payment Cost   |

Figure 2: Ablation study on heuristic search (w.r.t $D_\star^p \in [1, 10, 100]$). The results are averaged over 10 runs with standard deviation as the error bars.

## 5.2 Ablation Study on Heuristic Search

To investigate the impact of different components in our heuristic search, we compare the full-fledged model INC-FEDUCB with following variants on various environments: (1) INC-FEDUCB (w/o PF): without payment-free incentive mechanism, where the server only use money to incentivize clients; (2) INC-FEDUCB (w/o IS): without iterative search, where the server only rank the clients once. (3) INC-FEDUCB (w/o PF + IS): without both above strategies.

In Figure 2, we present the averaged learning trajectories of regret and communication cost, along with the final payment costs (normalized) under different $D_\star^p$. The results indicate that the full-fledged INC-FEDUCB consistently outperforms all other variants in various environments. Additionally, there is a substantial gap between the variants with and without the PF strategy, emphasizing the significance of leveraging the server's information advantage to motivate participation.

## 5.3 Environment & Hyper-Parameter Study

We further explored diverse $\beta$ hyper-parameter settings for INC-FEDUCB in various environments with varying $D_\star^p$, along with the comparison with DisLinUCB [40] (only comparable when $D_\star^p = 0$). Specifically, we explored different hyper-parameter settings for INC-FEDUCB with determinant ratio threshold $\beta \in [0.3, 0.7, 1]$, and various environmental configurations with data sharing cost $D_\star^p \in [1, 10, 100]$.

| $d = 25, K = 25$ | | DisLinUCB | INC-FEDUCB ($\beta = 1$) | INC-FEDUCB ($\beta = 0.7$) | INC-FEDUCB ($\beta = 0.3$) |
|---|---|---|---|---|---|
| | Regret (Acc.) | 48.46 | 48.46 | 48.46 ($\Delta = 0\%$) | 48.46 ($\Delta = 0\%$) |
| $T = 5,000, N = 50, D_\star^p = 0$ | Commu. Cost | 7,605,000 | 7,605,000 | 7,605,000 ($\Delta = 0\%$) | 7,605,000 ($\Delta = 0\%$) |
| | Pay. Cost | \ | 0 | 0 ($\Delta = 0\%$) | 0 ($\Delta = 0\%$) |
| | Regret (Acc.) | \ | 48.46 | 47.70 ($\Delta - 1.6\%$) | 48.38 ($\Delta - 0.2\%$) |
| $T = 5,000, N = 50, D_\star^p = 1$ | Commu. Cost | \ | 7,605,000 | 7,668,825 ($\Delta + 0.8\%$) | 7,733,575 ($\Delta + 1.7\%$) |
| | Pay. Cost | \ | 75.12 | 60.94 ($\Delta - 18.9\%$) | 22.34 ($\Delta - 70.3\%$) |
| | Regret (Acc.) | \ | 48.46 | 48.21 ($\Delta - 0.5\%$) | 47.55 ($\Delta - 1.9\%$) |
| $T = 5,000, N = 50, D_\star^p = 10$ | Commu. Cost | \ | 7,605,000 | 7,779,425 ($\Delta + 2.3\%$) | 8,599,950 ($\Delta + 13\%$) |
| | Pay. Cost | \ | 12,819.61 | 9,050.61 ($\Delta - 29.4\%$) | 4,859.17 ($\Delta - 62.1\%$) |
| | Regret (Acc.) | \ | 48.46 | 48.22 ($\Delta - 0.5\%$) | 48.44 ($\Delta - 0.1\%$) |
| $T = 5,000, N = 50, D_\star^p = 100$ | Commu. Cost | \ | 7,605,000 | 7,842,775 ($\Delta + 3.1\%$) | 8,718,425 ($\Delta + 14.6\%$) |
| | Pay. Cost | \ | 190,882.45 | 133,426.01 ($\Delta - 30.1\%$) | 88,893.78 ($\Delta - 53.4\%$) |

Table 1: Study on hyper-parameter of INC-FEDUCB and environment.

As shown in Table 1, when all clients are incentivized to share data, our INC-FEDUCB essentially recover the performance of DisLinUCB, while overcoming its limitation in incentivized settings when clients are not willing to share by default. Moreover, by reducing the threshold $\beta$, we can substantially save payment costs while still maintaining highly competitive regret, albeit at the expense of increased communication costs. And the reason for this increased communication cost has been explained before: more communication rounds will be triggered, as clients become more outdated.

## 6   Conclusion

In this work, we introduce a novel incentivized communication problem for federated bandits, where the server must incentivize clients for data sharing. We propose a general solution framework INC-FEDUCB, and initiate two specific implementations introducing data and monetary incentives, under the linear contextual bandit setting. We prove that INC-FEDUCB flexibly achieves customized levels of near-optimal regret with theoretical guarantees on communication and payment costs. Extensive empirical studies further confirmed our versatile designs in incentive search across diverse environments. Currently, we assume all clients truthfully reveal their costs of data sharing to the server. We are intrigued in extending our solution to settings where clients can exhibit strategic behaviors, such as misreporting their intrinsic costs of data sharing to increase their own utility. It is then necessary to study a truthful incentive mechanism design.

**Acknowledgement.**   We thank the anonymous reviewers for their insightful and constructive comments. This project is partially supported by NSF Award IIS-2213700 and IIS-2128019. Haifeng Xu is supported by an NSF Award CCF-2303372, an Army Research Office Award W911NF-23-1-0030, and an Office of Naval Research Award N00014-23-1-2802.

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

# A   Heuristic Search Algorithm

---

**Algorithm 4** Heuristic Search

---

**Require:** invalid client list $\{i_1, i_2, \cdots, i_k\}$, data-incentivized participant set $\widehat{S}_t$, and the last resort cost $\mathcal{I}_{\text{last}}^m$

1: Initialization: $S_t = \widehat{S}_t$
2: **for** $n \in [k]$ **do**
3:     Rank clients $\{i_1, \ldots, i_{k-n+1}\}$ (in new order) by Eq (6)
4:     $S_t = S_t \cup \{i_{k-n+1}\}$                              ▷ add the client with the largest contribution
5:     **for** client $j \in \{i_1, \ldots, i_{k-n}\}$ **do**          ▷ find extra data-incentivized participants
6:         Compute data incentive $\mathcal{I}_{j,t}^d$ for client $j$ by Eq (4)
7:         **if** $\mathcal{I}_{j,t}^d > D_j^p$ **then**
8:             $S_t = S_t \cup \{\Delta V_{j,t}\}$
9:     Compute total payment $\mathcal{I}_{n,t}^m = \sum_{i \in \widetilde{S}_t \setminus S_t} \mathcal{I}_{i,t}^m$ by Eq (5)
10:     **if** $\mathcal{I}_{n,t}^m \leq \mathcal{I}_{\text{last}}^m$ **then**
11:         **return** $S_t = \widehat{S}_t \cup \{\Delta V_{i_\alpha, t}\}$                      ▷ return *last resort*
12:     **else**
13:         **if** $\det(\Sigma(S_t) + V_{g,t-1}) / \det(\Sigma(\widetilde{S}_t) + V_{g,t-1}) > \beta$ **then**
14:             **return** $S_t$                              ▷ return *search result*

---

As sketched in Section 4.3, we devised an iterative search method based on the following ranking heuristic (formally defined in Eq. (6)): the more one client assists in increasing the server's determinant, the more valuable its contribution is, and thus we should motivate the most valuable clients to participate. Denote $n \leq k$ (initialized as 1) as the number of clients to be selected from the invalid list $\{i_1, \ldots, i_k\}$, and initialize the participant set $S_t = \widehat{S}_t$. In each round $n$, we rank the remaining $k - n + 1$ clients based on their potential contribution to the server by Eq. (6), and add the most valuable one to $S_t$ (Line 3-4). With the latest $S_t$ committed, we then proceed to determine additional data-incentivized participants by Eq. (4) (Line 5-8), and compute the total payment by Eq. (5) (Line 9). If having $n$ clients results in the total cost $\mathcal{I}_{n,t}^m > \mathcal{I}_{\text{last}}^m$, then we terminate the search and resort to our *last resort* (Line 10-11). Otherwise, if the resulting $S_t$ enables the server to satisfy the $\beta$ gap requirement, then we successfully find a better solution than *last resort* and can terminate the search. However, if having $n$ client is insufficient for the server to pass the $\beta$ gap requirement, we increase $n = n + 1$ and repeat the search process (Line 12-14). In particular, if the above process fails to terminate (i.e., having all $m$ clients still not suffices, we will still use the *last resort*. Note that, by utilizing matrix computation to calculate the contribution list in each round, this method only incurs a linear time complexity of $O(N)$, when $n = m = N$.

# B   Technical Lemmas

**Lemma 5 (Lemma H.3 of [40])** *With probability* $1 - \delta$, *single step pseudo-regret* $r_t = \langle \theta^*, \mathbf{x}^* - \mathbf{x}_t \rangle$ *is bounded by*

$$r_t \leq 2 \left( \sqrt{2 \log \left( \frac{\det(V_{i_t,t})^{1/2} \det(\lambda I)^{-1/2}}{\delta} \right)} + \lambda^{1/2} \right) \|\mathbf{x}_t\|_{V_{i_t,t}^{-1}} = O\left( \sqrt{d \log \frac{T}{\delta}} \right) \|\mathbf{x}_t\|_{V_{i_t,t}^{-1}} .$$

**Lemma 6 (Lemma 11 of [1])** *Let* $\{X_t\}_{t=1}^\infty$ *be a sequence in* $\mathbb{R}^d$, $V$ *is a* $d \times d$ *positive definite matrix and define* $V_t = V + \sum_{s=1}^t X_s X_s^\top$. *Then we have that*

$$\log \left( \frac{\det(V_n)}{\det(V)} \right) \leq \sum_{t=1}^n \|X_t\|_{V_{t-1}^{-1}}^2 .$$

*Further, if* $\|X_t\|_2 \leq L$ *for all* $t$, *then*

$$\sum_{t=1}^n \min \left\{ 1, \|X_t\|_{V_{t-1}^{-1}}^2 \right\} \leq 2 \left( \log \det(V_n) - \log \det V \right) \leq 2 \left( d \log \left( (\text{trace}(V) + nL^2)/d \right) - \log \det V \right).$$

**Lemma 7 (Lemma 12 of [1])** *Let $A$, $B$ and $C$ be positive semi-definite matrices such that $A = B + C$. Then, we have that*

$$\sup_{\mathbf{x} \neq \mathbf{0}} \frac{\mathbf{x}^\top A \mathbf{x}}{\mathbf{x}^\top B \mathbf{x}} \leq \frac{\det(A)}{\det(B)}.$$

## C Proof of Theorem 3

Our proof relies on the following lower bound result for federated linear bandits established in [13].

**Lemma 8 (Theorem 5.3 of [13])** *Let $p_i$ denote the probability that an agent $i \in [N]$ will communicate with the server at least once over time horizon $T$. Then for any algorithm with*

$$\sum_{i=1}^{N} p_i \leq \frac{c}{2C} \cdot \frac{N}{\log(T/N)} \tag{7}$$

*there always exists a linear bandit instance with $\sigma = L = S = 1$, such that for $T \geq Nd$, the expected regret of this algorithm is at least $\Omega(d\sqrt{NT})$.*

In the following, we will create a situation, where Eq. (7) always holds true for the payment-free incentive mechanism. Specifically, recall that the payment-free incentive mechanism (Section 4.2) motivates clients to participate using only data, i.e., the determinant ratio defined in Eq. (4) that indicates how much client $i$'s confidence ellipsoid can shrink using the data offered by the server. Based on matrix determinant lemma [6], we know that $\mathcal{I}_{i,t} \leq (1 + \frac{L^2}{\lambda})^T$. Additionally, by applying the determinant-trace inequality (Lemma 10 of [1]), we have $\mathcal{I}_{i,t} \leq (1 + \frac{TL^2}{\lambda d})^d$. Therefore, as long as $D_i^p > \min\{(1 + \frac{L^2}{\lambda})^T, (1 + \frac{TL^2}{\lambda d})^d\}$, where the tighter choice between the two upper bounds depends on the specific problem instance (i.e., either $d$ or $T$ being larger), it becomes impossible for the server to incentivize client $i$ to participate in the communication. Now based on Lemma 8, if the number of clients that satisfy $D_i^p \leq \min\{(1 + \frac{L^2}{\lambda})^T, (1 + \frac{TL^2}{\lambda d})^d\}$ is smaller than $\frac{c}{2C} \cdot \frac{N}{\log(T/N)}$, a sub-optimal regret of the order $\Omega(d\sqrt{NT})$ is inevitable for payment-free incentive mechanism, which finishes the proof. ∎

## D Proof of Theorem 4

To prove this theorem, we first need the following lemma.

**Lemma 9 (Communication Frequency Bound)** *By setting the communication threshold $D_c = \frac{T}{N^2 d \log T} - \sqrt{\frac{T^2}{N^2 dR \log T}} \log \beta$, the total number of epochs defined by the communication rounds satisfies,*

$$P = O(Nd \log T)$$

*where $R = \lceil d \log(1 + \frac{T}{\lambda d}) \rceil = O(d \log T)$.*

*Proof of Lemma 9.* Denote $P$ as the total number of epochs divided by communication rounds throughout the time horizon $T$, and $V_{g,t_p}$ as the aggregated covariance matrix at the $p$-th epoch. Specifically, $V_{g,t_0} = \lambda I$, $\widetilde{V}_T$ is the covariance matrix constructed by all data points available in the system at time step $T$.

Note that according to the incentivized communication scheme in INC-FEDUCB, not all clients will necessarily share their data in the last epoch, hence $\det(V_{g,t_P}) \leq \det(\widetilde{V}_T) \leq \left(\frac{tr(\widetilde{V}_T)}{d}\right) \leq (\lambda + T/d)^d$. Therefore,

$$\log \frac{\det(V_{g,t_P})}{\det(V_{g,t_{P-1}})} + \log \frac{\det(V_{g,t_{P-1}})}{\det(V_{g,t_{P-2}})} + \cdots + \log \frac{\det(V_{g,t_1})}{\det(V_{g,t_0})} = \log \frac{\det(V_{g,t_P})}{\det(V_{g,t_0})} \leq \left\lceil d \log(1 + \frac{T}{\lambda d}) \right\rceil$$

Let $\alpha \in \mathbb{R}^+$ be an arbitrary positive value, for epochs with length greater than $\alpha$, there are at most $\lceil \frac{T}{\alpha} \rceil$ of them. For epochs with length less than $\alpha$, say the $p$-th epoch triggered by client $i$, we have

$$\Delta t_{i,t_p} \cdot \log \frac{\det(V_{i,t_p})}{\det(V_{i,t_{\text{last}}})} > D_c$$

Combining the $\beta$ gap constraint defined in Section 4.3 and the fact that the server always downloads to all clients at every communication round, we have $\Delta t_{i,t_p} \leq \alpha$ and hence

$$\log \frac{\det(g, V_{t_p})}{\det(V_{g,t_{p-1}})} \geq \log \frac{\beta \cdot \det(\widetilde{V}_{t_p})}{\det(V_{g,t_{p-1}})} \geq \log \frac{\beta \cdot \det(V_{i,t_p})}{\det(V_{g,t_{p-1}})} \geq \log \frac{\beta \cdot \det(V_{i,t_p})}{\det(V_{i,t_{\text{last}}})} \geq \frac{D_c}{\alpha} + \log \beta$$

Let $R = \lceil d \log(1 + \frac{T}{\lambda d}) \rceil = O(d \log T)$, therefore, there are at most $\lceil \frac{R}{\frac{D_c}{\alpha} + \log \beta} \rceil$ epochs with length less than $\alpha$ time steps. As a result, the total number of epochs $P \leq \lceil \frac{T}{\alpha} \rceil + \lceil \frac{R}{\frac{D_c}{\alpha} + \log \beta} \rceil$. Note that $\lceil \frac{T}{\alpha} \rceil + \lceil \frac{R}{\frac{D_c}{\alpha} + \log \beta} \rceil \geq 2\sqrt{\frac{TR}{D_c + \alpha \log \beta}}$, where the equality holds when $\alpha = \sqrt{\frac{T(D_c + \alpha \log \beta)}{R}}$.

Furthermore, let $D_c = \frac{T}{N^2 d \log T} - \alpha \log \beta$, then $\alpha = \sqrt{\frac{T^2}{N^2 dR \log T}}$, we have

$$P = O\left(\sqrt{\frac{TR}{D_c + \alpha \log \beta}}\right) = O(N\sqrt{dR \log T}) = O(Nd \log T) \tag{8}$$

This concludes the proof of Lemma 9. ∎

**Communication Cost:** The proof of communication cost upper bound directly follows Lemma 9. In each epoch, all clients first upload $O(d^2)$ scalars to the server and then download $O(d^2)$ scalars. Therefore, the total communication cost is $C_T = P \cdot O(Nd^2) = O(N^2 d^3 \log T)$ ∎

**Monetary Incentive Cost:** Under the clients' committed data sharing cost $D^p = \{D_1^p, \cdots, D_N^p\}$, during each communication round at time step $t_p$, we only pay clients in the participant set $S_{t_p}$. Specifically, the payment (i.e., monetary incentive cost) $\mathcal{I}_{i,t_p}^m = 0$ if the data incentive is already sufficient to motivate the client to participate, i.e., when $\mathcal{I}_{i,t_p}^d \geq D_i^p$. Otherwise, we only need to pay the minimum amount of monetary incentive such that Eq. (1) is satisfied, i.e., $\mathcal{I}_{i,t_p}^m = D_i^p - \mathcal{I}_{i,t_p}^d$. Therefore, the accumulative monetary incentive cost is

$$\begin{aligned}
M_T = \sum_{p=1}^P \sum_{i=1}^N \mathcal{I}_{i,t_p}^m &= \sum_{p=1}^P \sum_{i=1}^N \max\left\{0, D_i^p - \mathcal{I}_{i,t_p}^d\right\} \cdot \mathbb{I}(\Delta V_{i,t_p} \in S_{t_p}) \\
&\leq \sum_{p=1}^P \sum_{i=1}^N \max\left\{0, \max_{i \in [N]}\{D_i^p\} - \mathcal{I}_{i,t_p}^d\right\} \cdot \mathbb{I}(\Delta V_{i,t_p} \in S_{t_p}) \\
&\leq \sum_{p=1}^P \sum_{i \in \bar{\mathcal{N}}_p} (\max_{i \in [N]}\{D_i^p\} - \mathcal{I}_{i,t_p}^d) \cdot \mathbb{I}(\Delta V_{i,t_p} \in S_{t_p}) \\
&\leq \max_{i \in [N]}\{D_i^p\} \sum_{p=1}^P \sum_{i=1}^N \mathbb{I}(\Delta V_{i,t_p} \in S_{t_p}) - \sum_{p=1}^P \sum_{i \in \bar{\mathcal{N}}_p} \mathcal{I}_{i,t_p}^d \cdot \mathbb{I}(\Delta V_{i,t_p} \in S_{t_p}) \\
&= \max_{i \in [N]}\{D_i^p\} \sum_{p=1}^P N_p - \sum_{i=1}^N \sum_{p \in \bar{\mathcal{P}}_i} \mathcal{I}_{i,t_p}^d
\end{aligned}$$

where $P$ and $N$ represent the number of epochs and clients, $N_p$ is the number of participants in $p$-th epoch, $\bar{\mathcal{N}}_p$ is the set of money-incentivized participants in the $p$-th epoch, $\bar{\mathcal{P}}_i$ is the set of epochs where client $i$ gets monetary incentive, whose size is denoted as $P_i = |\bar{\mathcal{P}}_i|$. Denote $D_{\max}^p = \max_{i \in [N]}\{D_i^p\}$ to simplify our later discussion.

Recall the definition of data incentive and $D_{i,t_p}(S_{t_p}) = \sum_{j:\{\Delta V_{j,t_p} \in S_{t_p}\} \wedge \{j \neq i\}} \Delta V_{j,t_p} + \Delta V_{-i,t_p}$ introduced in Eq. (4), we can show that

$$\begin{aligned}
\mathcal{I}_{i,t_p}^d &= \frac{\det\left(D_{i,t_p}(S_{t_p}) + V_{i,t_p}\right)}{\det(V_{i,t_p})} - 1 \\
&\geq \frac{\det(V_{g,t_p})}{\det(V_{i,t_p})} - 1
\end{aligned}$$

Therefore, we have

$$M_T \leq D_{\max}^p \cdot \sum_{p=1}^{P} N_p + \sum_{i=1}^{N} \sum_{p \in \bar{\mathcal{P}}_i} 1 - \sum_{i=1}^{N} \sum_{p \in \bar{\mathcal{P}}_i} \frac{\det(V_{g,t_p})}{\det(V_{i,t_p})}$$

$$\leq D_{\max}^p \cdot \sum_{p=1}^{P} N_p + \sum_{i=1}^{N} P_i - \sum_{i=1}^{N} P_i \cdot \left( \frac{\det(V_{g,t_1})}{\det(V_{i,t_1})} \cdot \frac{\det(V_{g,t_2})}{\det(V_{i,t_2})} \cdots \frac{\det(V_{g,t_{P_i}})}{\det(V_{i,t_{P_i}})} \right)^{\frac{1}{P_i}}$$

$$\leq D_{\max}^p \cdot \sum_{p=1}^{P} N_p + \sum_{i=1}^{N} P_i - \sum_{i=1}^{N} P_i \cdot \left( \frac{\det(V_{g,t_1})}{\det(V_{i,t_1})} \cdot \frac{\det(V_{i,t_1})}{\det(V_{i,t_2})} \cdots \frac{\det(V_{i,t_{P_i-1}})}{\det(V_{i,t_{P_i}})} \right)^{\frac{1}{P_i}}$$

$$= D_{\max}^p \cdot \sum_{p=1}^{P} N_p + \sum_{i=1}^{N} P_i - \sum_{i=1}^{N} P_i \cdot \left( \frac{\det(V_{g,t_1})}{\det(V_{i,t_{P_i}})} \right)^{\frac{1}{P_i}}$$

$$\leq (1 + D_{\max}^p) \cdot P \cdot N - \sum_{i=1}^{N} P_i \cdot \left( \frac{\det(\lambda I)}{\det(V_T)} \right)^{\frac{1}{P_i}}$$

where the second step holds by Cauchy-Schwarz inequality and the last step follows the facts that $P_i \leq P$, $N_p \leq N$, $\det(V_{g,t_1}) \geq \det(\lambda I)$, and $\det(V_{i,t_{P_i}}) \leq \det(V_T)$.

Specifically, by setting the communication threshold $D_c = \frac{T}{N^2 d \log T} - \sqrt{\frac{T^2}{N^2 dR \log T}} \log \beta$, where $R = \lceil d \log(1 + \frac{T}{\lambda d}) \rceil$, we have the total number of epochs $P = O(Nd \log T)$ (Lemma 9). Therefore,

$$M_T \leq (1 + D_{\max}^p) \cdot O(N^2 d \log T) - \sum_{i=1}^{N} P_i \cdot \left( \frac{\det(\lambda I)}{\det(V_T)} \right)^{\frac{1}{P_i}}$$

$$= O(N^2 d \log T)$$

which finishes the proof. ∎

**Regret:** To prove the regret upper bound, we first need the following lemma.

**Lemma 10 (Instantaneous Regret Bound)** *Under threshold $\beta$, with probability $1 - \delta$, the instantaneous pseudo-regret $r_t = \langle \theta^*, \mathbf{x}^* - \mathbf{x}_t \rangle$ in $j$-th epoch is bounded by*

$$r_t = O\left( \sqrt{d \log \frac{T}{\delta}} \right) \cdot \|\mathbf{x}_t\|_{\widetilde{V}_{t-1}^{-1}} \cdot \sqrt{\frac{1}{\beta} \cdot \frac{\det(V_{g,t_j})}{\det(V_{g,t_{j-1}})}}$$

*Proof of Lemma 10.* Denote $\widetilde{V}_t$ as the covariance matrix constructed by all available data in the system at time step $t$. As introduced in Eq. (3), the instantaneous regret of client $i$ is upper bounded by

$$r_t \leq 2\alpha_{i_t,t-1} \sqrt{\mathbf{x}_t^\top \widetilde{V}_{t-1}^{-1} \mathbf{x}_t} \cdot \sqrt{\frac{\det(\widetilde{V}_{t-1})}{\det(V_{i_t,t-1})}} = O\left( \sqrt{d \log \frac{T}{\delta}} \right) \cdot \|\mathbf{x}_t\|_{\widetilde{V}_{t-1}^{-1}} \cdot \sqrt{\frac{\det(\widetilde{V}_{t-1})}{\det(V_{i_t,t-1})}}$$

Suppose the client $i_t$ appears at the $j$-th epoch, i.e., $t_{j-1} \leq t \leq t_j$. As the server always downloads the aggregated data to every client in each communication round, we have

$$\frac{\det(\widetilde{V}_t)}{\det(V_{i_t,t})} \leq \frac{\det(\widetilde{V}_t)}{\det(V_{i_t,t_{j-1}})} \leq \frac{\det(\widetilde{V}_t)}{\det(V_{g,t_{j-1}})}$$

Combining the $\beta$ gap constraint defined in Section 4.3, we can show that

$$\frac{\det(\widetilde{V}_t)}{\det(V_{i_t,t})} \leq \frac{\det(\widetilde{V}_t)}{\det(V_{g,t_{j-1}})} \leq \frac{\det(V_{g,t_j})/\beta}{\det(V_{g,t_{j-1}})} = \frac{1}{\beta} \cdot \frac{\det(V_{g,t_j})}{\det(V_{g,t_{j-1}})}$$

Lastly, plugging the above inequality into Eq. (3), we have

$$r_t = O\left( \sqrt{d \log \frac{T}{\delta}} \right) \cdot \|\mathbf{x}_t\|_{\widetilde{V}_{t-1}^{-1}} \cdot \sqrt{\frac{1}{\beta} \cdot \frac{\det(V_{g,t_j})}{\det(V_{g,t_{j-1}})}}$$

which finishes the proof of Lemma 10. ∎

Now, we are ready to prove the accumulative regret upper bound. Similar to DisLinUCB [40], we group the communication epochs into *good epochs* and *bad epochs*.

**Good Epochs**: Note that for good epochs, we have $1 \leq \frac{\det(V_{g,t_j})}{\det(V_{g,t_{j-1}})} \leq 2$. Therefore, based on Lemma 10, the instantaneous regret in good epochs is

$$r_t = O\left(\sqrt{d \log \frac{T}{\delta}}\right) \cdot \|\mathbf{x}_t\|_{\widetilde{V}_{t-1}^{-1}} \cdot \sqrt{\frac{2}{\beta}}$$

Denote the accumulative regret among all good epochs as $REG_{good}$, then using the Cauchy–Schwarz inequality we can see that

$$
\begin{aligned}
REG_{good} &= \sum_{p \in P_{good}} \sum_{t \in \mathcal{B}_p} r_t \\
&\leq \sqrt{T \cdot \sum_{p \in P_{good}} \sum_{t \in \mathcal{B}_p} r_t^2} \\
&\leq O\left(\sqrt{T \cdot d \log \frac{T}{\delta} \cdot \frac{2}{\beta} \sum_{p \in P_{good}} \sum_{t \in \mathcal{B}_p} \|\mathbf{x}_t\|_{\widetilde{V}_{t-1}^{-1}}^2}\right)
\end{aligned}
$$

Combining the fact $x \leq 2 \log(1+x), \forall x \in [0, 1]$ and Lemma 6, we have

$$
\begin{aligned}
REG_{good} &\leq O\left(\sqrt{T \cdot \frac{d}{\beta} \log \frac{T}{\delta} \sum_{p \in P_{good}} \sum_{t \in \mathcal{B}_p} 2 \log\left(1 + \|\mathbf{x}_t\|_{\widetilde{V}_{t-1}^{-1}}^2\right)}\right) \\
&\leq O\left(\sqrt{T \cdot \frac{d}{\beta} \log \frac{T}{\delta} \cdot \sum_{p \in P_{good}} \log \frac{\det(\widetilde{V}_{t_p})}{\det(\widetilde{V}_{t_{p-1}})}}\right) \\
&\leq O\left(\sqrt{T \cdot \frac{d}{\beta} \log \frac{T}{\delta} \sum_{p \in P_{All}} \log \frac{\det(\widetilde{V}_{t_p})}{\det(\widetilde{V}_{t_{p-1}})}}\right) \\
&= O\left(\sqrt{T \cdot \frac{d}{\beta} \log \frac{T}{\delta} \cdot \log \frac{\det(\widetilde{V}_{t_P})}{\det(\widetilde{V}_{t_0})}}\right) \\
&\leq O\left(\sqrt{T \cdot \frac{d}{\beta} \log \frac{T}{\delta} \cdot d \log\left(1 + \frac{T}{\lambda d}\right)}\right) \\
&= O\left(\frac{d}{\sqrt{\beta}} \cdot \sqrt{T} \cdot \sqrt{\log \frac{T}{\delta} \cdot \log T}\right)
\end{aligned}
$$

**Bad Epochs**: Now moving on to the bad epoch. For any bad epoch starting from time step $t_s$ to time step $t_e$, the regret in this epoch is

$$REG = \sum_{t=t_s}^{t_e} r_t = \sum_{i=1}^{N} \sum_{\tau \in \mathcal{N}_i(t_e) \backslash \mathcal{N}_i(t_s)} r_\tau$$

where $\mathcal{N}_i(t) = \{1 \leq \tau \leq t : i_\tau = i\}$ denotes the set of time steps when client $i$ interacts with the environment up to $t$. Combining the fact $r_\tau \leq 2$ and Lemma 5, we have

$$r_\tau \leq \min\{2, 2\alpha_{i_\tau, \tau-1} \sqrt{\mathbf{x}_\tau^\top V_{i_\tau, \tau-1}^{-1} \mathbf{x}_\tau}\} = O\left(\sqrt{d \log \frac{T}{\delta}}\right) \min\{1, \|\mathbf{x}_\tau\|_{V_{i_\tau, \tau-1}^{-1}}\}$$

Therefore,

$$
\begin{aligned}
REG &\leq O\left(\sqrt{d\log\frac{T}{\delta}}\right)\sum_{i=1}^{N}\sum_{\tau\in\mathcal{N}_i(t_e)\backslash\mathcal{N}_i(t_s)}\min\{1,\|\mathbf{x}_\tau\|_{V_{i,\tau-1}^{-1}}\}\\
&\leq O\left(\sqrt{d\log\frac{T}{\delta}}\right)\sum_{i=1}^{N}\sqrt{\Delta t_{i,t_e}\sum_{\tau\in\mathcal{N}_i(t_e)\backslash\mathcal{N}_i(t_s)}\min\{1,\|\mathbf{x}_\tau\|_{V_{i,\tau-1}^{-1}}^2\}}\\
&\leq O\left(\sqrt{d\log\frac{T}{\delta}}\right)\sum_{i=1}^{N}\sqrt{\Delta t_{i,t_e}\sum_{\tau\in\mathcal{N}_i(t_e)\backslash\mathcal{N}_i(t_s)}\log\left(1+\|\mathbf{x}_\tau\|_{V_{i,\tau-1}^{-1}}^2\right)}\\
&= O\left(\sqrt{d\log\frac{T}{\delta}}\right)\sum_{i=1}^{N}\sqrt{\Delta t_{i,t_e}\sum_{\tau\in\mathcal{N}_i(t_e)\backslash\mathcal{N}_i(t_s)}\log\left(\frac{\det(V_{i,\tau})}{\det(V_{i,\tau-1})}\right)}\\
&\leq O\left(\sqrt{d\log\frac{T}{\delta}}\right)\sum_{i=1}^{N}\sqrt{\Delta t_{i,t_e}\cdot\log\frac{\det(V_{i,t_e})}{\det(V_{i,t_{\text{last}}})}}\\
&\leq O\left(\sqrt{d\log\frac{T}{\delta}}\right)N\cdot\sqrt{D_c}.
\end{aligned}
$$

where the second step holds by the Cauchy-Schwarz inequality, the third step follows from $x\leq 2\log(1+x),\forall x\in[0,1]$, the fourth step utilizes the elementary algebra, and the last two steps follow the fact that no client triggers the communication before $t_e$.

Recall that, as introduced in Lemma 9, the number of bad epochs is less than $R=\lceil d\log(1+\frac{T}{\delta})\rceil = O(d\log T)$, therefore the regret across all bad epochs is

$$
\begin{aligned}
REG_{bad} &= O\left(\sqrt{d\log\frac{T}{\delta}}\right)N\cdot\sqrt{D_c}\cdot O(d\log T)\\
&= O\left(Nd^{1.5}\sqrt{D_c\cdot\log\frac{T}{\delta}}\log T\right)
\end{aligned}
$$

Combining the regret for all good and bad epochs, we have accumulative regret

$$
\begin{aligned}
R_T &= REG_{good}+REG_{bad}\\
&= O\left(\frac{d}{\sqrt{\beta}}\cdot\sqrt{T}\cdot\sqrt{\log\frac{T}{\delta}\cdot\log T}\right)+O\left(Nd^{1.5}\sqrt{D_c\cdot\log\frac{T}{\delta}}\log T\right)
\end{aligned}
$$

According to Lemma 10, the above regret bound holds with high probability $1-\delta$. For completeness, we also present the regret when it fails to hold, which is bounded by $\delta\cdot\sum r_t\leq 2T\cdot\delta$ in expectation. And this can be trivially set to $O(1)$ by selecting $\delta=1/T$. In this way, we can primarily focus on analyzing the following regret when the bound holds.

$$
R_T = O\left(\frac{d}{\sqrt{\beta}}\sqrt{T}\log T\right)+O\left(Nd^{1.5}\log^{1.5}T\cdot\sqrt{D_c}\right)
$$

With our choice of $D_c=\frac{T}{N^2d\log T}-\sqrt{\frac{T^2}{N^2dR\log T}}\log\beta$ in Lemma 9, we have

$$
R_T = O\left(\frac{d}{\sqrt{\beta}}\sqrt{T}\log T\right)+O\left(Nd^{1.5}\log^{1.5}T\cdot\sqrt{\frac{T}{N^2d\log T}-\sqrt{\frac{T^2}{N^2dR\log T}}\log\beta}\right)
$$

Plugging in $R=\lceil d\log(1+\frac{T}{\lambda d})\rceil = O(d\log T)$, we get

$$
R_T = O\left(\frac{d}{\sqrt{\beta}}\sqrt{T}\log T\right)+O\left(Nd^{1.5}\log^{1.5}T\cdot\sqrt{\frac{T}{N^2d\log T}+\frac{T}{Nd\log T}\log\frac{1}{\beta}}\right)
$$

Furthermore, by setting $\beta > e^{-\frac{1}{N}}$, we can show that $\frac{T}{N^2 d \log T} > \frac{T}{N d \log T} \log \frac{1}{\beta}$, and therefore

$$R_T = O\left(\frac{d}{\sqrt{\beta}}\sqrt{T}\log T\right) + O\left(d\sqrt{T}\log T\right) = O\left(d\sqrt{T}\log T\right)$$

This concludes the proof. ∎

# E  Detailed Experimental Results

In addition to the empirical studies on the synthetic datasets reported in Section 5, we also conduct comprehensive experiments on the real-world recommendation dataset MovieLens [12]. Following [3, 22], we pre-processed the dataset to align it with the linear bandit problem setting, with feature dimension $d = 25$ and arm set size $K = 25$. Specifically, it contains $N = 54$ users and 26567 items (movies), where items receiving non-zero ratings are considered as having positive feedback, i.e., denoted by a reward of 1; otherwise, the reward is 0. In total, there are $T = 214729$ interactions, with each user having at least 3000 observations. By default, we set all clients' costs of data sharing as $D_i^p = D_\star^p \in \mathbb{R}, \forall i \in [N]$.

## E.1  Payment-free vs. Payment-efficient incentive mechanism (Supplement to Section 5.1)

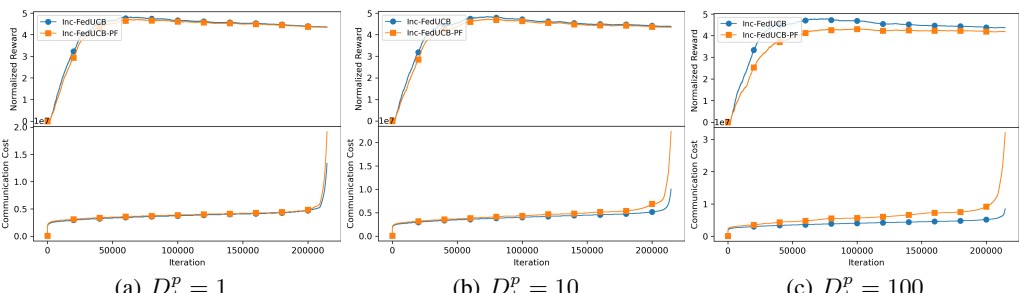

Figure 3: Comparison between payment-free vs. payment-efficient incentive designs.

Aligned with the findings presented in Section 5.1, the results on real-world dataset also confirm the advantage of the payment-efficient mechanism over the payment-free incentive mechanism in terms of both accumulative (normalized) reward and communication cost. As illustrated in Figure 3, this performance advantage is particularly notable in a more conservative environment, where clients have higher $D_\star^p$. And when the cost of data sharing for clients is relatively low, the performance gap between the two mechanisms becomes less significant. We attribute this to the fact that clients with low $D_i^p$ values are more readily motivated by the data alone, thus alleviating the need for additional monetary incentive. On the other hand, higher values of $D_i^p$ indicate that clients are more reluctant to share their data. As a result, the payment-free incentive mechanism fails to motivate a sufficient number of clients to participate in data sharing, leading to a noticeable performance gap, as evidenced in Figure 3(c). Note that the communication cost exhibits a sharp increase towards the end of the interactions. This is because the presence of highly skewed user distribution in the real-world dataset. For instance, in the last 2,032 rounds, only one client remains actively engaged with the environment, rapidly accumulating sufficient amounts of local updates, thus resulting in an increase in both communication frequency and cost.

## E.2  Ablation Study on Heuristic Search (Supplement to Section 5.2)

We further study the effectiveness of each component in the heuristic search of INC-FEDUCB on the real-world dataset and compare the performance among different variants across different data sharing costs. As presented in Figure 4(a) and 4(b), the variants without payment-free (PF) component, which only rely on monetary incentive to motivate clients to participate, generally exhibit lower rewards and higher communication costs. The reason is a bit subtle: as the payment efficient mechanism is subject to both $\beta$ gap constraint and minimum payment cost requirement, it tends to satisfy the $\beta$ gap constraint with minimum amount of data collected. But the payment free mechanism will always collect the maximum amount data possible. As a result, without the PF component, the server

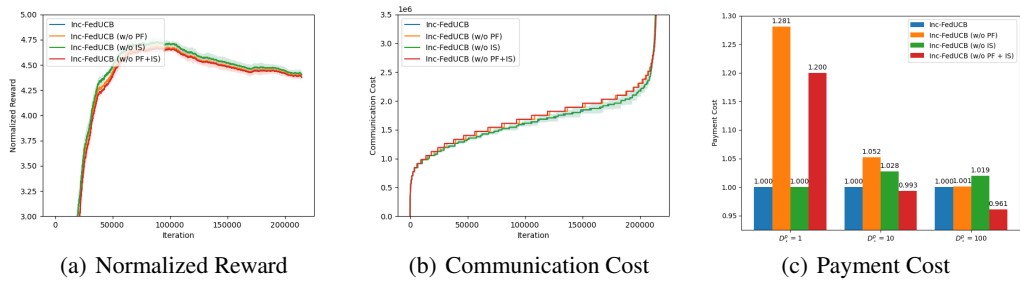

(a) Normalized Reward        (b) Communication Cost        (c) Payment Cost

Figure 4: Ablation study on heuristic search (w.r.t $D_\star^p \in [1, 10, 100]$).

tends to collect less (but enough) data, which in turn leads to more communication rounds and worse regret. The side-effect of the increased communication frequency is the higher payment costs, with respect to the $\beta$ gap requirement in each communication round. This is particularly notable in a more collaborative environment, where clients have lower data sharing costs. As exemplified in Figure 4(c), when the clients are more willing to share data (e.g., $D_\star^p = 1$), the variants without PF incur significantly higher payment costs compared to the those with PF, as the server misses the opportunity to get those easy to motivate clients. Therefore, providing data incentives becomes even more crucial in such scenarios to ensure effective client participation and minimize payment costs. On the other hand, the variants without iterative search (IS) tend to maintain competitive performance compared to the fully-fledged model, despite incurring a higher payment cost, highlighting the advantage of IS in minimizing payment.

### E.3 Environment & Hyper-Parameter Study (Supplement to Section 5.3)

| $d = 25, K = 25, D_c = \frac{T}{N^2 d \log T} - \sqrt{\frac{T^2}{N^2 dR \log T}} \cdot \log \beta$ | | DisLinUCB | Inc-FedUCB ($\beta = 1$) | Inc-FedUCB ($\beta = 0.7$) | Inc-FedUCB ($\beta = 0.3$) |
|---|---|---|---|---|---|
| MovieLens ($D_\star^p = 0$) | Reward (Acc.) | 38,353 | 38,353 | 37,731 ($\Delta - 1.6\%$) | 36,829 ($\Delta - 2.4\%$) |
| | Commu. Cost | 33,415,200 | 33,415,200 | 5,967,000 ($\Delta - 82\%$) | 2,457,000 ($\Delta - 92.6\%$) |
| | Pay. Cost | \ | 0 | 0 ($\Delta = 0\%$) | 0 ($\Delta = 0\%$) |
| MovieLens ($D_\star^p = 1$) | Reward (Acc.) | \ | 38,353 | 37,717 ($\Delta - 1.7\%$) | 36,833 ($\Delta - 4\%$) |
| | Commu. Cost | \ | 33,415,200 | 13,372,250 ($\Delta - 60\%$) | 5,038,675 ($\Delta - 84.9\%$) |
| | Pay. Cost | \ | 7859.67 | 124.41 ($\Delta - 98.4\%$) | 0 ($\Delta - 100\%$) |
| MovieLens ($D_\star^p = 10$) | Reward (Acc.) | \ | 38,353 | 37,648 ($\Delta - 1.8\%$) | 36,675 ($\Delta - 4.4\%$) |
| | Commu. Cost | \ | 33,415,200 | 10,041,250 ($\Delta - 70\%$) | 4,240,625 ($\Delta - 87.3\%$) |
| | Pay. Cost | \ | 110,737.62 | 8,590.43 ($\Delta - 92.2\%$) | 2,076.98 ($\Delta - 98.1\%$) |
| MovieLens ($D_\star^p = 100$) | Regret (Acc.) | \ | 38,353 | 37,641 ($\Delta - 1.9\%$) | 36,562 ($\Delta - 4.7\%$) |
| | Commu. Cost | \ | 33,415,200 | 8,496,600 ($\Delta - 74.6\%$) | 5,136,700 ($\Delta - 84.6\%$) |
| | Pay. Cost | \ | 1,155,616.99 | 105,847.84 ($\Delta - 90.8\%$) | 32,618.34 ($\Delta - 97.2\%$) |

Table 2: Study on hyper-parameter of INC-FEDUCB and environment (w/ theoretical $D_c$).

In contrast to the hyper-parameter study on synthetic dataset with fixed communication threshold reported in Section 5.3, in this section, we comprehensively investigate the impact of $\beta$ and $D_\star^p$ on the real-world dataset by varying the communication thresholds $D_c$. First, we empirically validate the effectiveness of the theoretical value of $D_c = \frac{T}{N^2 d \log T} - \sqrt{\frac{T^2}{N^2 dR \log T}}$ as introduced in Theoreom 4. The results presented in Table 2 are generally consistent with the findings in Section 5.3: decreasing $\beta$ can substantially lower the payment cost while still maintaining competitive rewards. We can also find that using the theoretical value of $D_c$ can also save the communication cost. This results from the fact that setting $D_c$ as a function of $\beta$ leads to a higher communication threshold for lower $\beta$, and therefore reducing communication frequency. This observation is essentially aligned with the intuition behind lower $\beta$: when the system has a higher tolerance for outdated sufficient statistics, it should not only pay less in each communication round but also trigger communication less frequently.

On the other hand, we investigate INC-FEDUCB's performance under two fixed communication thresholds $D_c = T/(N^2 d \log T)$ and $D_c = T/(N d \log T)$, which are presented in Table 3 and 4,

| $d = 25, K = 25, D_c = \frac{T}{N^2 d \log T}$ | | DisLinUCB | INC-FEDUCB ($\beta = 1$) | INC-FEDUCB ($\beta = 0.7$) | INC-FEDUCB ($\beta = 0.3$) |
|---|---|---|---|---|---|
| MovieLens ($D_\star^p = 0$) | Reward (Acc.) | 38,353 | 38,353 | 38,353 ($\Delta = 0\%$) | 38,353 ($\Delta = 0\%$) |
| | Commu. Cost | 33,415,200 | 33,415,200 | 33,415,200 ($\Delta = 0\%$) | 33,415,200 ($\Delta = 0\%$) |
| | Pay. Cost | \ | 0 | 0 ($\Delta = 0\%$) | 0 ($\Delta = 0\%$) |
| MovieLens ($D_\star^p = 1$) | Reward (Acc.) | \ | 38,353 | 38,207 ($\Delta - 0.4\%$) | 38,208 ($\Delta - 0.4\%$) |
| | Commu. Cost | \ | 33,415,200 | 171,046,600 ($\Delta + 412\%$) | 191,280,875 ($\Delta + 472\%$) |
| | Pay. Cost | \ | 7859.67 | 2095.73 ($\Delta - 73.3\%$) | 36.32 ($\Delta - 99.5\%$) |
| MovieLens ($D_\star^p = 10$) | Reward (Acc.) | \ | 38,353 | 38,251 ($\Delta - 0.3\%$) | 37,609 ($\Delta - 1.9\%$) |
| | Commu. Cost | \ | 33,415,200 | 135,521,025 ($\Delta + 306\%$) | 424,465,650 ($\Delta + 1170\%$) |
| | Pay. Cost | \ | 110,737.62 | 33,271.39 ($\Delta - 70\%$) | 33,872.78 ($\Delta - 69.4\%$) |
| MovieLens ($D_\star^p = 100$) | Reward (Acc.) | \ | 38,353 | 38,251 ($\Delta - 0.3\%$) | 37,970 ($\Delta - 1\%$) |
| | Commu. Cost | \ | 33,415,200 | 135,521,025 ($\Delta + 306\%$) | 522,196,225 ($\Delta + 1463\%$) |
| | Pay. Cost | \ | 1,155,616.99 | 352,231.39 ($\Delta - 69.5\%$) | 346,619.77 ($\Delta - 70\%$) |

Table 3: Study on hyper-parameter of INC-FEDUCB and environment (w/ lower fixed $D_c$).

| $d = 25, K = 25, D_c = \frac{T}{N d \log T}$ | | DisLinUCB | INC-FEDUCB ($\beta = 1$) | INC-FEDUCB ($\beta = 0.7$) | INC-FEDUCB ($\beta = 0.3$) |
|---|---|---|---|---|---|
| MovieLens ($D_\star^p = 0$) | Reward (Acc.) | 37,308 | 37,308 | 37,308 ($\Delta = 0\%$) | 37,308 ($\Delta = 0\%$) |
| | Commu. Cost | 2,737,800 | 2,737,800 | 2,737,800 ($\Delta = 0\%$) | 2,737,800 ($\Delta = 0\%$) |
| | Pay. Cost | \ | 0 | 0 ($\Delta = 0\%$) | 0 ($\Delta = 0\%$) |
| MovieLens ($D_\star^p = 1$) | Reward (Acc.) | \ | 37,308 | 37,296 ($\Delta - 0.1\%$) | 37,306 ($\Delta - 0.1\%$) |
| | Commu. Cost | \ | 2,737,800 | 4,197,525 ($\Delta + 53.3\%$) | 5,948,950 ($\Delta + 117.3\%$) |
| | Pay. Cost | \ | 55.31 | 44.76 ($\Delta - 19.1\%$) | 0 ($\Delta - 100\%$) |
| MovieLens ($D_\star^p = 10$) | Reward (Acc.) | \ | 37,308 | 37,297 ($\Delta - 0.1\%$) | 37,167 ($\Delta - 0.1\%$) |
| | Commu. Cost | \ | 2,737,800 | 3,696,350 ($\Delta + 35\%$) | 5,765,075 ($\Delta + 110.6\%$) |
| | Pay. Cost | \ | 4048.69 | 3779.77 ($\Delta - 6.6\%$) | 2242.22 ($\Delta - 44.6\%$) |
| MovieLens ($D_\star^p = 100$) | Reward (Acc.) | \ | 37,308 | 37,273 ($\Delta - 0.1\%$) | 36,946 ($\Delta - 0.1\%$) |
| | Commu. Cost | \ | 2,737,800 | 3,484,850 ($\Delta + 27.3\%$) | 5,690,250 ($\Delta + 107.8\%$) |
| | Pay. Cost | \ | 77,041.04 | 65,286.90 ($\Delta - 15.3\%$) | 40,010.59 ($\Delta - 48.1\%$) |

Table 4: Study on hyper-parameter of INC-FEDUCB and environment (w/ higher fixed $D_c$).

respectively. These two values are created by increasing the theoretical value of $D_c$. Overall, the main findings align with those reported in Section 5.3, confirming our previous statements. While reducing $\beta$ can achieve competitive rewards with lower payment costs, it comes at the expense of increased communication costs, suggesting the trade-off between payment costs and communication costs. Interestingly, the setting under a higher $D_\star^p$ and $D_c$ can help mitigate the impact of $\beta$. Specifically, while increasing the client's cost of data sharing inherently brings additional incentive costs, raising the communication threshold results in fewer communication rounds, leading to reduced overall communication costs. This finding highlights the importance of thoughtful design in choosing $D_c$ and $\beta$ to balance the trade-off between payment costs and communication costs in real-world scenarios with diverse data sharing costs.

## E.4 Extreme Case Study

To further investigate the utility of INC-FEDUCB in extreme cases, we conduct a set of case studies on both synthetic and real-world datasets with fixed data sharing costs. As shown in Table 5 and 6, when $\beta$ is extremely small, we can achieve almost $100\%$ savings in incentive cost compared to the case where every client has to be incentivized to participate in data sharing (i.e., $\beta = 1$). However, this extreme setting inevitably results in a considerable drop in regret/reward performance and potentially tremendous extra communication cost due to the extremely outdated local statistics in clients. Nevertheless, by strategically choosing the communication threshold, we can mitigate the additional communication costs associated with the low $\beta$ values. For instance, in the synthetic dataset, the difference in performance drop between the theoretical $D_c$ setting and heuristic $D_c$ value

| $d = 25, K = 25, D^p_* = 100$ | | Inc-FedUCB ($\beta = 1$) | Inc-FedUCB ($\beta = 0.7$) | Inc-FedUCB ($\beta = 0.3$) | Inc-FedUCB ($\beta = 0.01$) |
|---|---|---|---|---|---|
| $T = 5000, N = 50$ $(D_c = T/N^2 d \log T)$ | Regret (Acc.) | 45.37 | 46.33 ($\Delta + 2.1\%$) | 48.49 ($\Delta + 6.9\%$) | 51.22 ($\Delta + 12.9\%$) |
| | Commu. Cost | 174,720,000 | 264,193,275 ($\Delta + 51.2\%$) | 299,134,900 ($\Delta + 71.2\%$) | 314,667,500 ($\Delta + 80.1\%$) |
| | Pay. Cost | 479,397.18 | 229,999.66 ($\Delta - 52\%$) | 115,600 ($\Delta - 75.9\%$) | 42,800 ($\Delta - 91.1\%$) |
| $T = 5000, N = 50$ $(D_c = T/N^2 d \log T - \sqrt{T^2/N^2 dR \log T} \cdot \log \beta)$ | Regret (Acc.) | 45.37 | 46.72 ($\Delta + 3\%$) | 49.13 ($\Delta + 8.3\%$) | 53.72 ($\Delta + 18.4\%$) |
| | Commu. Cost | 174,720,000 | 17,808,725 ($\Delta - 89.8\%$) | 7,237,600 ($\Delta - 95.9\%$) | 2,981,175 ($\Delta - 98.3\%$) |
| | Pay. Cost | 479,397.18 | 178,895.78 ($\Delta - 62.7\%$) | 84,989.39 ($\Delta - 82.3\%$) | 1,200 ($\Delta - 99.7\%$) |

Table 5: Case study on synthetic dataset.

| $d = 25, K = 25, D^p_* = 100$ | | Inc-FedUCB ($\beta = 1$) | Inc-FedUCB ($\beta = 0.7$) | Inc-FedUCB ($\beta = 0.3$) | Inc-FedUCB ($\beta = 0.01$) |
|---|---|---|---|---|---|
| MovieLens $(D_c = T/N^2 d \log T)$ | Reward (Acc.) | 38,353 | 38,251 ($\Delta - 0.3\%$) | 37,970 ($\Delta - 1\%$) | 37,039 ($\Delta - 3.4\%$) |
| | Commu. Cost | 33,415,200 | 135,521,025 ($\Delta + 306\%$) | 522,196,225 ($\Delta + 1463\%$) | 1,226,741,425 ($\Delta + 3571.2\%$) |
| | Pay. Cost | 1,155,616.99 | 352,231.39 ($\Delta - 69.5\%$) | 346,619.77 ($\Delta - 70\%$) | 75,799.39 ($\Delta - 93.4\%$) |
| MovieLens $(D_c = T/N^2 d \log T - \sqrt{T^2/N^2 dR \log T} \cdot \log \beta)$ | Reward (Acc.) | 38,353 | 37,641 ($\Delta - 1.9\%$) | 36,562 ($\Delta - 4.7\%$) | 31,873 ($\Delta - 16.9\%$) |
| | Commu. Cost | 33,415,200 | 8,496,600 ($\Delta - 74.6\%$) | 5,136,700 ($\Delta - 84.6\%$) | 1,880,450 ($\Delta - 94.4\%$) |
| | Pay. Cost | 1,155,616.99 | 105,847.84 ($\Delta - 90.8\%$) | 32,618.34 ($\Delta - 97.2\%$) | 200 ($\Delta - 99.9\%$) |

Table 6: Case study on real-world dataset.

is relatively small ($\Delta + 18.4\%$ vs. $\Delta + 12.9\%$). However, these two different choices of $D_c$ exhibit opposite effects on communication costs, with the theoretical one achieving a significant reduction ($\Delta - 98.3\%$) while the heuristic one incurred a significant increase ($\Delta + 80.1\%$). On the other hand, in the real-world dataset, the heuristic choice of $D_c$ may lead to a smaller performance drop compared to the theoretical setting of $D_c$ (e.g., $\Delta - 3.4\%$ vs. $\Delta - 16.9\%$), reflecting the specific characteristics of the environment (e.g., a high demand of up-to-date sufficient statistics). Similar to the findings in Section E.3, this case study also emphasizes the significance of properly setting the system hyper-parameter $\beta$ and $D_c$. By doing so, we can effectively accommodate the trade-off between performance, incentive costs, and communication costs, even in extreme cases.

## F   Notation Table

| Notation | Meaning |
|---|---|
| $d$ | context dimension |
| $N$ | total number of clients |
| $T$ | total number of time steps |
| $\beta$ | hyperparameter that controls the regret level |
| $D_c$ | communication threshold |
| $D^p_i$ | data sharing cost of client $i$ |
| $S_t$ | participant set at time step $t$ |
| $D_{i,t}(S_t)$ | data offered by the sever to client $i$ at time step $t$ |
| $\mathcal{I}^d_{i,t}/\mathcal{I}^m_{i,t}$ | data/monetary incentive for client $i$ at time step $t$ |
| $\widetilde{V}_t$ | covariance matrix constructed by all available data in the system |
| $V_{i,t}, b_{i,t}$ | local data of client $i$ at time step $t$ |
| $V_{g,t}, b_{g,t}$ | global data stored at the server at time step $t$ |
| $\Delta V_{i,t}, \Delta b_{i,t}$ | data stored at client $i$ that has not been shared with the server |
| $\Delta V_{-i,t}, \Delta b_{-i,t}$ | data stored at the server that has not been shared with the client $i$ |

Table 7: Main technical notations used in this paper.

