# OpenReview forum: "Incentivized Communication for Federated Bandits"
_NeurIPS.cc/2023/Conference — NeurIPS 2023 poster_

### Official Review · Reviewer_U6am · 2023-07-01

**Soundness:** 3 good
**Presentation:** 2 fair
**Contribution:** 3 good
**Rating:** 5
**Confidence:** 3

**Summary:**

The authors study a new problem in federated bandits that involves incentivizing clients to share data. They propose a solution called Inc-FedUCB, which offers incentives in a linear contextual bandit setting. They demonstrate that Inc-FedUCB can achieve near-optimal regret levels with guarantees on communication and payment costs. They also conduct extensive experiments to validate the effectiveness of their incentive designs in various environments.

**Strengths:**

1. The paper is the first work that formulates and proves theoretical guarantees for the incentive design in federated bandit learning.
2. Designing incentives to encourage collaborations among clients is important in federated bandit learning.
3. The proposed algorithms are supported theoretically and numerically.

**Weaknesses:**

The paper's focus is on designing incentivized communication protocols. The contribution is meaningful but as mentioned by the authors in Line 51, a well-defined metric to measure the utility of data sharing is important. The current presentation related to the utility design is not clear. For example,
  - Line 126: Did the authors assume that all the clients share the same $\theta_{\star}$? If true, the assumption might be strong.
  - Eqn. (4): It seems that the value of new data is independent of $\theta_{\star}$. Again, does this need the assumption that all clients share the same $\theta_{\star}$? If not, the value should also depend on different $\theta$ because some high-value data for client 1 may be useless for other clients.
 - Lemmas 5 and 7 seem to be important in designing the incentive but are deferred to Appendix.

**Questions:**

Please address the questions in weakness.

---

> ### Author Rebuttal · Authors · 2023-08-08
>
> We thank the reviewer for the constructive suggestions to clarify problem formulation and data valuation design, as well as for helping us improve the overall organization of the paper.
>
>
> **[Q1]**: Line 126: Did the authors assume that all the clients share the same $\theta_\star$? If true, the assumption might be strong.
>
> **[A1]**: Thanks for pointing out the place that could cause unnecessary confusion. We want to clarify that in our work all the clients share the same $\theta_\star$, and this is actually the standard assumption and widely adopted in the federated bandits literature [1,2,3,4]. For a detailed discussion of our data utility design, please refer to our response to CQ2 in the general feedback provided to all reviewers.
>
> **[Q2]**: Eqn. (4): It seems that the value of new data is independent of $\theta_\star$. Again, does this need the assumption that all clients share the same $\theta_\star$? If not, the value should also depend on different $\theta$ because some high-value data for client 1 may be useless for other clients.
>
> **[A2]**: As the first work of this new incentivized federated bandit problem, we start with the standard homogeneous clients setting [1,2,3,4], where clients share the same unknown reward parameter $\theta_\star$. In this way, the data valuation design does **NOT** require to be dependent on $\theta_\star$ since all clients are estimating the same $\theta_\star$, and this design aligns well with the client’s objective of regret minimization as explained in Section 4.2.
>
> We absolutely agree that it is interesting to explore the heterogeneous setting where clients are associated with different $\theta_\star$. And depending on the problem assumption, our current data valuation design of Eq(4) may or may not apply to the heterogeneous setting. Please see a detailed answer to CQ2 in the general response to all reviewers.
>
>
> **[Q3]**: Lemmas 5 and 7 seem to be important in designing the incentive but are deferred to Appendix.
>
> **[A3]**: Thanks for the great suggestion! Due to space limit, after presenting our original contribution in the main paper, we had to make such compromises and leave existing important lemmas to the appendix. In the updated version (w/ more space), we will ensure better organization of the paper and address this concern adequately.
>
> **References**
>
> [1] Yuanhao Wang, Jiachen Hu, Xiaoyu Chen, and Liwei Wang. Distributed bandit learning: Near-optimal regret with efficient communication. ICLR 2020
>
> [2] Ruiquan Huang, Weiqiang Wu, Jing Yang, and Cong Shen. Federated linear contextual bandits. NeurIPS 2021.
>
> [3] Chuanhao Li and Hongning Wang. Asynchronous upper confidence bound algorithms for federated linear bandits. AISTATS 2022.
>
> [4] Chuanhao Li, Huazheng Wang, Mengdi Wang, and Hongning Wang. Learning kernelized contextual bandits in a distributed and asynchronous environment. ICLR 2023.

---

> > ### Comment · Reviewer_U6am · 2023-08-14
> >
> > Thanks for the rebuttal. I keep my original rating.

---

> > > ### Author Response · Authors · 2023-08-15
> > >
> > > We sincerely appreciate the reviewer’s constructive feedback on our submission, and we are glad that the reviewer is satisfied with our submission and rebuttal. As the first work investigating and facilitating incentives in federated bandit research, we are very excited to share our results and findings in improving the efficiency and practical operability of federated bandit learning with the community, from a fresh yet realistic perspective that every client must be motivated to participate. With this in mind, we are truly enthusiastic about integrating any additional suggestions the reviewer might have that would further improve our work and increase the chance of publication of our work.
> > >
> > > We look forward to the possibility of your updated evaluation, which will undoubtedly contribute to the advancement of our work and open up an interesting field in this line of research.

---

### Official Review · Reviewer_NbUe · 2023-07-05

**Soundness:** 3 good
**Presentation:** 2 fair
**Contribution:** 4 excellent
**Rating:** 5
**Confidence:** 2

**Summary:**

This paper introduces a novel federated learning protocol, so that 1) the setting is online instead of the more common offline setting, and 2) during each iteration, each client only chooses to participate (sharing information with the central server) if the client is gaining a sufficient amount of utility via participation through an incentive mechanism. The second property is particularly interesting because in the traditional setting, every client unconditionally participates and exchanges information with the central server, while in reality they may be reluctant because of low potential benefits. To ensure sufficient number of participants during every iteration, the central server may also provide extra support to motivate some clients. Finally, the protocol obtains near-optimal regrets and reasonable communication and incentive costs.

**Strengths:**

1. The topic is important and the mechanism based upon monetary and non-monetary incentives is novel.

2. The paper is generally well-written, with convincing support for the motivation. It also covers an extensive literature as references to earlier works in this area.

3. Such incentive mechanism may be well applied to other related areas.

4. The theoretical argument is extensive. However, please take a look in the weaknesses and questions section.

**Weaknesses:**

1. We may need a more organized guide to notations in such a paper with a large number of variables. In particular, I didn’t find the definition of $g$ in $V_{g,0}$ and $b_{g,0}$ in the line 1 of algorithm 1. The same is in line 15: what do the negative subscripts $-j$ mean?

2. The bar of utility for the participation standard did not have sufficient support: in line 9 of algorithm 1, why are the determinants of those matrices a good standard for a client to decide whether to participate or not?

**Questions:**

The questions are listed in the weaknesses section.

**Limitations:**

I did not find any concerns in this regard.

---

> ### Author Rebuttal · Authors · 2023-08-09
>
> We thank the reviewer for the constructive suggestions to clarify data valuation design and improve the organization of notations.
>
> **[Q1]**: We may need a more organized guide to notations in such a paper with a large number of variables. In particular, I didn’t find the definition of $g$ in $V_{g,0}$ and $b_{g,0}$ in the line 1 of algorithm 1. The same is in line 15: what do the negative subscripts $-j$ mean?
>
> **[A1]**: Thanks for the great suggestion! Following the standard notation in the literature [1], $V_{g,t}, b_{g,t}$ represents the global (g) sufficient statistics stored in the server at time step $t$. The negative subscripts $V_{-j, t}, b_{-j, t}$ represent the aggregated updates stored at the server that have not been sent to client $j$. As suggested by the reviewer, we have added a notation table for the main technical notations. Please refer to the supplementary PDF for rebuttal (in our general response to all reviewers).
>
> **[Q2]**: The bar of utility for the participation standard did not have sufficient support: in line 9 of algorithm 1, why are the determinants of those matrices a good standard for a client to decide whether to participate or not?
>
> **[A2]**: We should clarify the confusion. A client decides to participate only if the incentive offered by the server exceeds its cost, as defined in Eq(1). Line 9 of algorithm 1 does **NOT** imply the decision of participation but rather serves as an event trigger for communication (see detailed description at Line 201 in Section 4.1). As highlighted in Algorithm 1, once a communication event is triggered, all clients will upload their $\Delta V_{i,t}$ to the server (Line 10 of Algorithm 1) so as to compute potential incentives for clients. After that, the client will upload its corresponding $\Delta b_{i,t}$ only if the incentive offered by the server is deemed to exceed its data sharing cost (Line 13 of Algorithm 1). Note that this design does not compromise clients’ privacy because only having $V_{i,t}$ is insufficient to update the model and the clients’ secret essentially lies in $\Delta b_{i,t}$, as explained in Section 4.1 (Line 212). For a further discussion on the data valuation design, please find our answer to CQ2 in the general response to all reviewers.
>
> **References**
>
> [1] Chuanhao Li and Hongning Wang. Asynchronous upper confidence bound algorithms for federated linear bandits. AISTATS 2022.

---

> > ### Comment · Reviewer_NbUe · 2023-08-16
> >
> > Thanks for the response and justification, which solve my concerns. I keep the original rating.

---

> > > ### Author Response · Authors · 2023-08-17
> > >
> > > It is great to know that our responses properly addressed the reviewer's concerns, and we are grateful for the reviewer’s constructive feedback and favorable evaluation of our submission. Being the first work to investigate and facilitate incentives in federated bandit research, we are very excited to share our results and findings in this new direction with the community, which we believe will significantly enhance the efficiency and practical operability of federated bandit learning. Therefore, we are truly enthusiastic about integrating any additional suggestions the reviewer might have that would further improve our work and increase the chances of publication.
> > >
> > > We look forward to the possibility of your updated evaluation, which will undoubtedly contribute to the advancement of our work and open up an interesting field in this line of research.

---

### Official Review · Reviewer_rmwW · 2023-07-07

**Soundness:** 3 good
**Presentation:** 3 good
**Contribution:** 4 excellent
**Rating:** 5
**Confidence:** 4

**Summary:**

The paper studied the problem of incentivizing data sharing in federated learning under the linear contextual federated bandit model with self-interested clients. While most previous works in federated bandit assume that all clients are willing participants in model sharing, this assumption is often unrealistic and neglects the inherent cost of data sharing for each client. The author proposed a general framework INC-FEDUCB that achieves near-optimal regret and provided upper bounds on the monetary and communication cost. Finally, the author provided empirical experiments to evaluate their mechanisms in different environments.

**Strengths:**

- The studied problem of incentives in federated bandit learning is interesting and novel.
- The theoretical claims are strong contributions.
- The payment-efficient mechanism, while using a heuristic method, provides an improvement over the naive method.
- In general, the paper is easy to read.
- The provided empirical experiments support the theoretical claims.

**Weaknesses:**

- The paper made an important assumption that the server knows the vector of cost values exactly as an input to the incentive mechanism. This assumption relies on clients truthfully reporting their personal costs, which can be leveraged by adversarial clients who purposefully misreport their costs in order to game the incentive system. Also, since an objective of federated learning is to protect the privacy of participating clients, relying on clients to disclose their personal costs seems unrealistic.

- Figures 1 and 2 do not have error bars.

- There are some minor typos in the paper.

- The monetary incentives might also come with some negative societal consequences, where clients are ranked by their potential contributions. There could be a scenario where clients are not selected due to their background and raise a fairness issue (e.g.: less-fortunate hospitals in a federated network that have fewer samples and thus do not contribute much are also less likely to be paid).

**Questions:**

- Can the current assumption on the cost function be relaxed by making a more explicit assumption on the form of the cost function for each client that account for the data-generating cost, communication cost, privacy cost, etc?

**Limitations:**

- The authors have addressed the limitation of the public cost vector. However, the author did not address the potential fairness issue that can come up with their monetary payment mechanism.

---

> ### Author Rebuttal · Authors · 2023-08-09
>
> We thank the reviewer for the constructive suggestions to clarify the important problem formulations, improve the presentation of results, and strengthen the discussion on the broader societal impact on fairness.
>
> **[Q1]**: The paper made an important assumption that the server knows the vector of cost values exactly as an input to the incentive mechanism. This assumption relies on clients truthfully reporting their personal costs, which can be leveraged by adversarial clients who purposefully misreport their costs in order to game the incentive system. Also, since an objective of federated learning is to protect the privacy of participating clients, relying on clients to disclose their personal costs seems unrealistic.
>
> **[A1]**: Thanks for pointing out the place that could cause unnecessary confusion. Please find our answer to CQ1 in the general response to all reviewers.
>
> **[Q2]**: Figures 1 and 2 do not have error bars.
>
> **[A2]**: Thanks for the great suggestion! We have updated the figures with error bars. Please refer to the supplementary PDF for rebuttal.
>
> **[Q3]**: There are some minor typos in the paper.
>
> **[A3]**: Thanks for your careful review! We have thoroughly examined the paper and corrected the typos in the updated version.
>
> **[Q4]**: The monetary incentives might also come with some negative societal consequences, where clients are ranked by their potential contributions. There could be a scenario where clients are not selected due to their background and raise a fairness issue (e.g.: less-fortunate hospitals in a federated network that have fewer samples and thus do not contribute much are also less likely to be paid).
>
> **[A4]**: Thanks for the valuable comments! We would like to clarify that in the federated bandit problems, the clients’ primary goal is to minimize their regret [1,2,3,4]. According to our Algorithm 1, all clients (no matter whether they get paid or not) will receive the same amount of data after each communication for better arm selection (thus minimizing regret). Therefore, there is **NO** fairness issue in the sense of regret minimization.  In fact, our incentive design is reasonably fair for both sides of the problem: 1. From a rational individual’s perspective, only those who suffer from data sharing (e.g., potential cost of privacy) will get monetary compensation; 2. From the system designer’s perspective, all clients are helped equally by the system in terms of regret minimization.
>
> Following the reviewer’s hospital example, a client (hospital) minimizing its regret can be interpreted as providing better treatment to its patients. Based on our incentive mechanism design, all hospitals will receive the same improvement of treatment quality by our federated bandit learning solution, therefore ensuring fairness in terms of helping patients. Although it may result in different hospital revenues, as some hospitals get more payments from the server, revenue distribution is beyond the scope of our work.
>
> With that being said, we absolutely agree fairness could be a concern in some application problems, e.g., when maximizing monetary utility is also part of the client’s objective. Additional treatment is then needed from the system side for fairness in this regard, e.g., providing extra monetary incentive even when data incentive is enough to motivate the client for participation. As also mentioned in the ethical review, enforcing fairness is undoubtedly an important issue in not only our problem setup, but more generally in modern machine learning. The present work primarily focuses on developing methods to incentivize communications for more efficient federated learning. We plan to add a discussion on potential fairness issues our methods may potentially cause to some specific applications, though developing systematic methodologies to address this issue seems out of the scope of our work.
>
> **[Q5]**: Can the current assumption on the cost function be relaxed by making a more explicit assumption on the form of the cost function for each client that account for the data-generating cost, communication cost, privacy cost, etc?
>
> **[A5]**: Thanks for the insightful suggestion! We would like to clarify that our current cost function is not limited to any specific cost forms, such as communication resource consumption, data production cost, or potential privacy loss. Instead, it is a general design that jointly considers multiple aspects and is represented by a scalar (as introduced in Section 3.2). As the reviewer suggested, we completely agree that it would be more flexible to further customize the cost function by making assumptions on fine-grained cost forms, especially when our method is applied to different scenarios where certain aspects may be prioritized over others. For example, study the case where the data sharing cost for each client is based on the amount of local data it possesses or cost varies by different clients’ productivity. Unlike the fixed cost setting, this dynamic cost may introduce more difficulties in incentive cost analysis, and we believe it's a worthwhile future direction to explore more cost function designs.
>
> **[Q6]**: The authors have addressed the limitation of the public cost vector. However, the author did not address the potential fairness issue that can come up with their monetary payment mechanism.
>
> **[A6]**: Please find our answer to Q4 in the above response.
>
> **References**
>
> [1] Yuanhao Wang, Jiachen Hu, Xiaoyu Chen, and Liwei Wang. Distributed bandit learning: Near-optimal regret with efficient communication. ICLR 2020
>
> [2] Ruiquan Huang, Weiqiang Wu, Jing Yang, and Cong Shen. Federated linear contextual bandits. NeurIPS 2021.
>
> [3] Chuanhao Li and Hongning Wang. Asynchronous upper confidence bound algorithms for federated linear bandits. AISTATS 2022.
>
> [4] Chuanhao Li, Huazheng Wang, Mengdi Wang, and Hongning Wang. Learning kernelized contextual bandits in a distributed and asynchronous environment. ICLR 2023.

---

> > ### Comment · Area_Chair_mAL9 · 2023-08-18
> >
> > Dear Reviewer rmwW,
> >
> > We would appreciate it if you could acknowledge the author's rebuttal and see if they have addressed your concerns. Thank you!
> >
> > AC

---

> > ### Comment · Reviewer_rmwW · 2023-08-18
> >
> > Thank you for the response and justification. I have updated my rating.

---

> > > ### Author Response · Authors · 2023-08-18
> > >
> > > We would like to express our gratitude to the reviewer for the positive response and the updated rating. As the first work to investigate and facilitate incentives in federated bandit research, we are very excited to share the results and findings of our research in this new direction with the community. And we firmly believe this will greatly enhance the efficiency and practicality of federated bandit learning.
> > >
> > > Therefore, we are more than happy to incorporate any additional suggestions the reviewer might have that could further enhance our work and lead to a more favorable evaluation. We have no doubt that the reviewer's valuable advocacy will significantly contribute to the advancement of our work and open up an interesting field in this line of research.

---

### Official Review · Reviewer_zWH8 · 2023-07-07

**Soundness:** 3 good
**Presentation:** 3 good
**Contribution:** 2 fair
**Rating:** 7
**Confidence:** 2

**Summary:**

This paper introduces an incentivized communication problem for federated bandits. They study the contextual linear bandit setting and propose the first incentivized communication protocol, namely, INC-FEDUCB, that achieves near-optimal regret with provable communication and incentive cost guarantees.

**Strengths:**

The paper is well-organized. The problem studied in the paper is novel and important. They provide both technical results and empirical experiments on both synthetic and real-world datasets.

**Weaknesses:**

The paper lacks intuitive explanations for the technical results.

**Questions:**

It might be great to add some high-level idea of the proofs for the theorems.

**Limitations:**

They assume all clients truthfully reveal their costs of data sharing to the server.

---

> ### Author Rebuttal · Authors · 2023-08-09
>
> We thank the reviewer for the appreciation of our work, and the constructive suggestions to enhance the presentation of our theoretical results.
>
> **[Q1]**: The paper lacks intuitive explanations for the technical results. It might be great to add some high-level idea of the proofs for the theorems.
>
>
> **[A1]**: Thanks for pointing out the places where we can make our paper more friendly to readers with diverse backgrounds. Below, we comprehensively summarize the intuition behind our algorithm design, as well as providing explanations of the main theoretical results.
>
> **Payment-free vs., Payment-efficient Mechanism Design**:
> As we explained in Section 4.2, maximizing the determinant of the client’s local data $V_{i, t}$ directly corresponds to reducing its regret in this federated bandit problem. This principle drives the design of our metric defined in Eq(4): the greater the server’s offer (denoted as $D_{i,t}(S_t)$) can increase the determinant of the client’s local data, the higher incentive it can provide. But as we proved in Theorem 3, the payment-free mechanism might not motivate any client to participate under specific circumstances. To address this issue, the payment-efficient incentive mechanism introduces additional monetary incentives to motivate clients. And to avoid trivially paying everyone more than enough, we look for minimum incentive cost to achieve the desired level of regret, controlled by the hyperparameter $\beta$. This poses a challenging optimization problem, as a brute-force search solution can yield a time complexity up to $O(2^N)$; and we implement a heuristic-based search (Algorithm 3) to minimize the incentive cost, with a time complexity of only $O(N)$.
>
> **Theorem 3**:  As detailed in Appendix C, the data incentive is bounded by the environment configuration. Thus, when client $i$’s cost $D^p_i$ exceeds this bound, it becomes impossible for the server to provide enough incentive to motivate client $i$ to participate in the communication. Theorem 3 implies that when the number of participating clients is less than the threshold $\frac{c}{2C} \cdot \frac{N}{\log(T/N)}$, a sub-optimal regret of the order $\Omega(d\sqrt{NT})$) (i.e., no regret reduction) is inevitable.
>
>
> The proof of Theorem 3 relies on Lemma 8, which indicates that once we encounter a situation where the number of participants in the payment-free mechanism falls below a threshold, a sub-optimal regret is inevitable. Therefore, we first establish a data incentive bound, and then create a situation specified in Lemma 8, which completes the proof.
>
> **Communication Cost**: As detailed in Appendix D (Line 519), the communication cost is $C_T = P \cdot O(Nd^2) = O(N^2 d^3 \log T)$, where $O(Nd^2)$ is the communication cost per epoch, and $P = O(Nd\log T)$ is the number of communication epochs, given the communication threshold $D_c = \frac{T}{N^2d\log T} - \sqrt{\frac{T^2}{N^2dR\log T} }\log \beta$. This result suggests that a lower value for $\beta$ results in a higher communication threshold $D_c$, thus leading to reduced communication frequency and cost, which is also supported by our numerical experiments (Appendix E.3).
>
> To bound the communication cost, we first analyze the communication cost per epoch, then establish an upper bound for the total number of epochs, thereby completing the proof.
>
>
> **Regret**: As explained in Appendix D (Line 567), the regret is analyzed under good epochs and bad epochs of communication. Our result $R_T = R_{good} + R_{bad} = O\left ( \frac{d }{\sqrt{\beta}}\cdot \sqrt{T} \cdot \sqrt{\log\frac{T}{\delta} \cdot \log T}\right) + O\left(Nd^{1.5}\sqrt{D_c \cdot \log\frac{T}{\delta}}\log T \right)$ shows that a larger value of $\beta$ leads to smaller regret in good epochs (first term), as the client’s local data gets closer to the global oracle. Meanwhile, a larger communication threshold $D_c$ results in worse regret in bad epochs (second term). This is because a higher communication threshold causes less frequent communications, ultimately resulting in worse regret.
>
> **Incentive Cost**: As detailed in Appendix D (Line 527), the incentive cost $M_T \leq \max_{i\in[N]}\{D_i^p\} \sum\limits_{p = 1}^P N_p - \sum\limits_{i=1}^N\sum\limits_{p \in \mathcal{\bar{P_i}}} \mathcal{I}^{d}_{i, t_p}$ consists of two parts. The first part represents the case where the server only uses money to motivate clients in each communication epoch, disregarding the data incentive. The second part represents how much monetary incentive the server could have saved by motivating clients with data incentive. With the proper $D_c$, a lower value of $\beta$ not only leads to less communication frequency but also decreases the demand for the server to collect data from the clients in each epoch, jointly resulting in reduced incentive cost, which is verified in our numerical experiments (Appendix E.3).
>
> To derive this bound, we first analyze the monetary incentive cost by associating it with the client’s data sharing cost and the data incentive already provided by the server. Then, by establishing a lower bound for the data incentive, we can upper bound the incentive cost.
>
> **[Q2]**: They assume all clients truthfully reveal their costs of data sharing to the server.
>
> **[A2]**: Please find our answer to CQ1 in the general response to all reviewers.

---

> > ### Comment · Reviewer_zWH8 · 2023-08-16
> >
> > Thanks for the quick response and explanation. I keep the original rating.

---

> > > ### Author Response · Authors · 2023-08-17
> > >
> > > We are glad that the reviewer found our responses helpful, and we genuinely thank the reviewer for recognizing our work. Indeed, we are very excited to introduce this novel incentivized setting into federated bandit research and share our results and findings in this new direction with the community.
> > >
> > > With the reviewer's valuable advocacy, we firmly believe that our work will pioneer an interesting and important field in this line of research, significantly enhancing the efficiency and practical operability of federated bandit learning.

---

### Author Rebuttal · Authors · 2023-08-09

# General response to the reviewers:

We sincerely thank all the reviewers for their thoughtful comments and constructive suggestions, which will significantly help us strengthen our paper. It is encouraging that all reviewers appreciate the novelty and importance of the studied problem and our proposed solution, with solid theoretical analysis (Reviewer zWH8, rmwW, NbUe, U6am), extensive numerical validation (Reviewer zWH8, rmwW, NbUe, U6am), and potential broader impact to other related areas where incentivized federated learning is needed (Reviewer NbUe, U6am).

There are also shared comments regarding truthful cost revealing and data utility design. Indeed, one of the intended goals of our work is to inspire further investigations into this new direction with more diverse settings, such as learning with strategic clients where truthful mechanism design is needed, and heterogeneous client settings where relevant data utility design is needed. And we agree with the reviewers that those are all important future directions. Next, we first provide our responses to these common questions (CQs), and endeavor to provide individual responses to each reviewer.

**[CQ1]**: Truthful cost revealing (Reviewer zWH8, rmwW)

**[CA1]**:  In this work, we introduce the incentivized communication problem for federated bandits. As the first work of its kind, we aim to establish a foundation and initiate the study with a simplified setting, where clients truthfully reveal their data sharing costs. But as we discussed in the conclusion section, the study of truthful mechanism design is a very interesting and important future work. Specifically, to prevent clients from strategically misreporting their costs to gain more utility (either monetary or data incentive) from the server, one potentially promising direction is to investigate truthful (also known as incentive-compatible) mechanisms like the Vickrey–Clarke–Groves (VCG) mechanism, under which being truthful is the best response for all clients. We firmly believe this will open up an interesting new field of studies in federated bandits and beyond.

On the other hand, as Reviewer rmwW suggested, explicitly sharing individual costs may expose a side channel of privacy breaches. In practice, one promising solution is to avoid this direct revelation via secure computation. As the cost values are simple scalars, like prices, and the associated operations are simply comparisons, it will not incur high overhead in secure computation.

**[CQ2]**: Data utility design (Reviewer NbUe, U6am)

**[CA2]**: In federated bandit problems, the clients typically share the same unknown reward parameter $\theta_\star$, which is a standard setting in this line of research [1,2,3,4]. We followed this setting and refer to it as the homogeneous client setting. As illustrated in Eq(3) of our paper, the determinant ratio $\frac{\det(\widetilde{V}\_{t-1})}{\det(V_{i_t, t-1})}$ reflects the additional regret due to the delayed synchronization between client $i_t$’s local sufficient statistics $V_{i_t, t-1}$ and the global statistics $\widetilde{V}_{t-1}$. This argument has been recognized in prior works, e.g., Section 3.2 of [3], and is also theoretically supported by Lemma 5 and 7 of our paper. In other words, if all clients participate in data sharing in every time step, the ratio will be kept at 1, and thus every client essentially enjoys the optimal regret (discussed in Section 3.1). **Therefore, minimizing this ratio directly corresponds to reducing client $i_t$'s regret**. As a result, the proposed Eq(4) is a natural design of data valuation for homogeneous clients. In essence, if we denote the data valuation function as $f(x)$, where $x$ is the metric defined in Eq(4) that directly measures the value of data for regret reduction, our current design can be regarded as $f(x)=x$. For different application scenarios, we can further generalize it to any monotonic function $f(x)$ on top of this metric.

Furthermore, we should emphasize that Line 9 of Algorithm 1 is **NOT** where the clients decide whether to participate. Instead, it is the “communication trigger” - to ensure communication efficiency during the federated learning process. An event trigger is introduced to control the communication frequency as detailed in Section 4.1.

We also acknowledge that there are also studies [3] that explore the heterogeneous client setting in federated bandits, where the unknown parameter $\theta_\star$ for each client consists of a globally shared component $\theta_\star^g$ and a unique local component $\theta_\star^l$. In this case, the data valuation design should be different from our current solution, because data valuation now depends on each client’s own $\theta_\star$, which is unknown to the server and clients. For example, we may need to assume the availability of additional knowledge about the relation among the set $\{\theta_\star\}$, or design strategies that estimate such relations on the fly. Alternatively, one simplified solution is to assume clients value the data only based on the shared component $\theta_\star^g$, and then the data valuation design will be essentially the same as our current choice.

**References**

[1] Yuanhao Wang, Jiachen Hu, Xiaoyu Chen, and Liwei Wang. Distributed bandit learning: Near-optimal regret with efficient communication. ICLR 2020

[2] Ruiquan Huang, Weiqiang Wu, Jing Yang, and Cong Shen. Federated linear contextual bandits. NeurIPS 2021.

[3] Chuanhao Li and Hongning Wang. Asynchronous upper confidence bound algorithms for federated linear bandits. AISTATS 2022.

[4] Chuanhao Li, Huazheng Wang, Mengdi Wang, and Hongning Wang. Learning kernelized contextual bandits in a distributed and asynchronous environment. ICLR 2023.

---

### Decision · Program_Chairs · 2023-09-21

**Decision:**

Accept (poster)

**Comment:**

This paper studies the problem of incentives in federated contextual linear bandits. The authors formalize the incentive problem and propose the corresponding mechanism to solve it. The authors also provide the regret guarantees and communication costs of the proposed method. This is a problem relevant for NeurIPS that has not been thoroughly studied before.

The reviewers are positive and agree that the problem and results are interesting. In addition, the authors were able to address the concerns of the reviewers in the discussion.

In light of this, I recommend the acceptance of the paper and encourage the authors to incorporate these discussions into the revision.